Novel Systems Biology Techniques

# Integrated Metabolic Modeling, Culturing, and Transcriptomics Explain Enhanced Virulence of *Vibrio cholerae* during Coinfection with Enterotoxigenic *Escherichia coli*

Alyaa M. Abdel-Haleem,[a,b] Vaishnavi Ravikumar,[c] Boyang Ji,[d] Katsuhiko Mineta,[a] Xin Gao,[a] Jens Nielsen,[c,d] Takashi Gojobori,[a,b] Ivan Mijakovic[c,d]

[a]King Abdullah University of Science and Technology (KAUST), Computational Bioscience Research Centre (CBRC), Thuwal, Saudi Arabia
[b]King Abdullah University of Science and Technology (KAUST), Biological and Environmental Sciences and Engineering (BESE) Division, Thuwal, Saudi Arabia
[c]Novo Nordisk Foundation Center for Biosustainability, Technical University of Denmark, Kongens Lyngby, Denmark
[d]Systems and Synthetic Biology, Department of Chemical and Biological Engineering, Chalmers University of Technology, Gothenburg, Sweden

**ABSTRACT** Gene essentiality is altered during polymicrobial infections. Nevertheless, most studies rely on single-species infections to assess pathogen gene essentiality. Here, we use genome-scale metabolic models (GEMs) to explore the effect of coinfection of the diarrheagenic pathogen *Vibrio cholerae* with another enteric pathogen, enterotoxigenic *Escherichia coli* (ETEC). Model predictions showed that *V. cholerae* metabolic capabilities were increased due to ample cross-feeding opportunities enabled by ETEC. This is in line with increased severity of cholera symptoms known to occur in patients with dual infections by the two pathogens. *In vitro* coculture systems confirmed that *V. cholerae* growth is enhanced in cocultures relative to single cultures. Further, expression levels of several *V. cholerae* metabolic genes were significantly perturbed as shown by dual RNA sequencing (RNAseq) analysis of its cocultures with different ETEC strains. A decrease in ETEC growth was also observed, probably mediated by nonmetabolic factors. Single gene essentiality analysis predicted conditionally independent genes that are essential for the pathogen's growth in both single-infection and coinfection scenarios. Our results reveal growth differences that are of relevance to drug targeting and efficiency in polymicrobial infections.

**IMPORTANCE** Most studies proposing new strategies to manage and treat infections have been largely focused on identifying druggable targets that can inhibit a pathogen's growth when it is the single cause of infection. *In vivo*, however, infections can be caused by multiple species. This is important to take into account when attempting to develop or use current antibacterials since their efficacy can change significantly between single infections and coinfections. In this study, we used genome-scale metabolic models (GEMs) to interrogate the growth capabilities of *Vibrio cholerae* in single infections and coinfections with enterotoxigenic *E. coli* (ETEC), which cooccur in a large fraction of diarrheagenic patients. Coinfection model predictions showed that *V. cholerae* growth capabilities are enhanced in the presence of ETEC relative to *V. cholerae* single infection, through cross-fed metabolites made available to *V. cholerae* by ETEC. *In vitro*, cocultures of the two enteric pathogens further confirmed model predictions showing an increased growth of *V. cholerae* in coculture relative to *V. cholerae* single cultures while ETEC growth was suppressed. Dual RNAseq analysis of the cocultures also confirmed that the transcriptome of *V. cholerae* was distinct during coinfection compared to single-infection scenarios where processes related to metabolism were significantly perturbed. Further, *in silico* gene-knockout simulations uncovered discrepancies in gene essentiality for *V. cholerae* growth between single infections and coinfections. Integrative model-guided analysis thus identified druggable targets that would be critical for *V.*

Address correspondence to Takashi Gojobori, takashi.gojobori@kaust.edu.sa, or Ivan Mijakovic, ivan.mijakovic@chalmers.se.

Single-gene essentiality analysis predicted conditionally independent genes that are essential for the pathogen's growth in both single-infection and coinfection scenarios.

*cholerae* growth in both single infections and coinfections; thus, designing inhibitors against those targets would provide a broader spectrum of coverage against cholera infections.

**KEYWORDS** infectious diseases, cholera, diarrhea, coinfection, drug target, flux balance analysis, constraint-based model, genome-scale reconstruction, *Vibrio* cholera, computational modeling, genome-scale modeling

Many studies focus on single-species infections although pathogens often cause infections as part of multispecies communities (1). Most studies that aim at identifying essential genomes, for example, have largely depended on single cultures (2–5). Such studies thus identify sets of "conditionally dependent essential" genes depending on the investigated growth conditions. Coinfecting microorganisms alter pathogen gene essentiality during polymicrobial infections (1). Nevertheless, a limited number of studies have attempted to identify variations in growth capabilities or gene essentiality of a pathogen under coinfection conditions.

Many metabolic processes are critical for cellular growth and survival, and hence a pathogen's anabolic and catabolic capabilities are usually tightly linked to its growth capabilities. There is growing evidence that, in addition to signals from the environment, the metabolism of a pathogen plays a major role in its virulence as well (6–9).

Genome-scale metabolic network reconstructions (GENREs) (10–12) have proven to be powerful tools to probe the metabolic capabilities of several enteric pathogens including *Escherichia coli* (13), *Shigella* (13), and *Salmonella* (14). GENREs are knowledge bases describing metabolic capabilities and the biochemical basis for entire organisms (10–12). GENREs can be mathematically formalized and combined with numerical representations of biological constraints and objectives to create genome-scale metabolic models (GEMs) (10–12). These GEMs can be used to predict biological outcomes (e.g., gene essentiality, growth rate) given an environmental context (e.g., metabolite availability [14, 15]). Metabolic models recapitulate the biological processes of nutrient uptake and metabolite secretion, which can be the basis of some microbial interactions (16). A growing number of experiments illustrated the predictive power of metabolic-driven computational approaches to describe emergent behaviors of coexisting species (17–22). However, deploying computational models to predict variations in pathogens' growth capabilities when present in single-infecting or coinfecting scenarios has not been investigated.

*Vibrio cholerae* is a Gram-negative bacterium that causes acute voluminous diarrhea representing a dramatic example of an enteropathogenic invasion. Cholera infections are typically caused by contaminated food and water (23, 24). Seven cholera pandemics have been recorded in modern history, and the latest is still ongoing (25–27). The *V. cholerae* life cycle is marked by repetitive transitions between aquatic environments and the host gastrointestinal tract; thus, it has to adjust to different qualities and quantities of nutrient sources (25). Within the human host, a highly active metabolic program is necessary to support *V. cholerae* high growth rates (25), where it was reported that cell numbers reach up to $10^9$ cells/g stool excreted by cholera patients (23, 25, 26). Further, several reports have suggested a role for central metabolism in regulating the production of virulence factors in *V. cholerae* (cholera toxin [CTX] and toxin-coregulated pilus [TCP]). For instance, TCP and CTX are not produced when *V. cholerae* is grown in M9-glycerol (27–29). The Entner-Doudoroff pathway has been shown to be obligatory for gluconate utilization and plays an important role in regulating *V. cholerae* virulence (29). While most case reports focus on *V. cholerae* as the single causative agent of diarrhea in cases of cholera infections, *V. cholerae* has commonly been involved in dual infections with enterotoxigenic *E. coli* (ETEC) (30–32), the second most frequent cause (~15%) of diarrheal diseases after *V. cholerae*. Notably, dual infections with *V. cholerae* and ETEC are associated with increased severity and increased health care costs (31). Thus, there is a need to study the variations in growth capabilities and gene essentiality between single- and multispecies infections of pathogens in general, and of *V. cholerae* in particular.

Here, we built a *V. cholerae* genome-scale metabolic model and validated its single gene essentiality predictions against experimentally published data. We then evaluated the growth capabilities of *V. cholerae* in relation to other enteric pathogens by simulating their growth under 656 growth conditions spanning several nutrient sources under aerobic and anaerobic conditions. Following that, we reconstructed a coinfection model of *V. cholerae* with ETEC in a shared environment and compared the growth capabilities of *V. cholerae* in single versus coinfection settings. Coinfection model simulations allowed for a comprehensive assessment of variations in growth capabilities and single gene essentiality when *V. cholerae* is grown solely or in coculture with ETEC. *In vitro* cocultures of the two enteric pathogens as well as dual transcriptome sequencing (RNAseq) data reflected corresponding variations in growth predictions and gene expression levels, respectively. Using single-infection and coinfection models, we predicted *V. cholerae* essential genes representing potential druggable targets that would be broader in spectrum against both *V. cholerae* single and coinfections. The present work is computationally driven using high-quality experimentally verified *in silico* and *in vitro* models and can be viewed as a means to prioritize potential druggable targets of pathogens that are known to be involved in single and multispecies infections. Further, our results substantiate the notion that data-driven computational modeling coupled to experiments can predict and analyze microbial communities' behavior.

## RESULTS

**Characterizing the metabolic capabilities of *V. cholerae*.** *i*AM-Vc960, a manually curated and quality-controlled GEM of *V. cholerae*, was constructed (Fig. 1, step 1) to probe the enteric pathogen's metabolic capabilities and gene essentiality in single infections and coinfections. We sequenced and annotated the genome of *V. cholerae* 52, an O37 serotype strain (see Materials and Methods and see also Fig. S1 at https:// github.com/alyamahmoud/coinfection_modeling/blob/master/supplementary _material/supplementary_text.docx). A list of metabolic pathways in *V. cholerae* V52 was built based on the genome annotation generated in this study as well as those available in PATRIC and that of *V. cholerae* O1 N16961 (see Table S1 at https://github .com/alyamahmoud/coinfection_modeling/blob/master/supplementary_material/ supplementary_tables.xlsx). The reconstruction was converted into a model, and the stoichiometric matrix was constructed with mass- and charge-balanced reactions in the standard fashion using the COBRA toolbox v.3.0 (33). Flux balance analysis (FBA) was used to assess network characteristics and perform simulations (34). The biomass function was constructed primarily based on that of *Vibrio vulnificus* (7) and *E. coli* K-12 *i*JO1366 (35). Transcriptomics data of *V. cholerae* V52 single cultures in minimal medium were also generated and used to further refine *i*AM-Vc960 reconstruction and biomass objective function (see Table S1 at https://github.com/alyamahmoud/coinfection _modeling/blob/master/supplementary_material/supplementary_tables.xlsx). *i*AM-Vc960 accounts for 2,172 reactions, 1,741 metabolites across three compartments (cytosol, periplasm, and extracellular compartments), and 960 metabolic genes. Gene-protein-reaction (GPR) associations could be defined for 72% of all enzymatic reactions (Fig. 2A). *i*AM-Vc960 exceeds the automatically generated *V. cholerae* model as part of the Path2Models (36) project in terms of its gene, metabolite, and reaction content. Five hundred eighty-four (89%) of the Path2Models *V. cholerae* model genes were already in *i*AM-Vc960. The remaining 68 genes were mostly nonmetabolic. The Path2Models *V. cholerae* model as downloaded from the biomodels repository was unable to produce any biomass; thus, we could not perform a functional comparison between *i*AM-Vc960 and the previously published *V. cholerae* model (see supplementary text at https:// github.com/alyamahmoud/coinfection_modeling/blob/master/supplementary _material/supplementary_text.docx for details on comparison to other previously published *V. cholerae* GEMs [37]).

The *i*AM-Vc960 predicted growth rate was 1.07 mmol/g (dry weight [DW])/h, in M9 minimal medium supplemented with glucose, corresponding to a doubling time of 39 min. Previous experiments (38) using *V. cholerae* species reported doubling times of

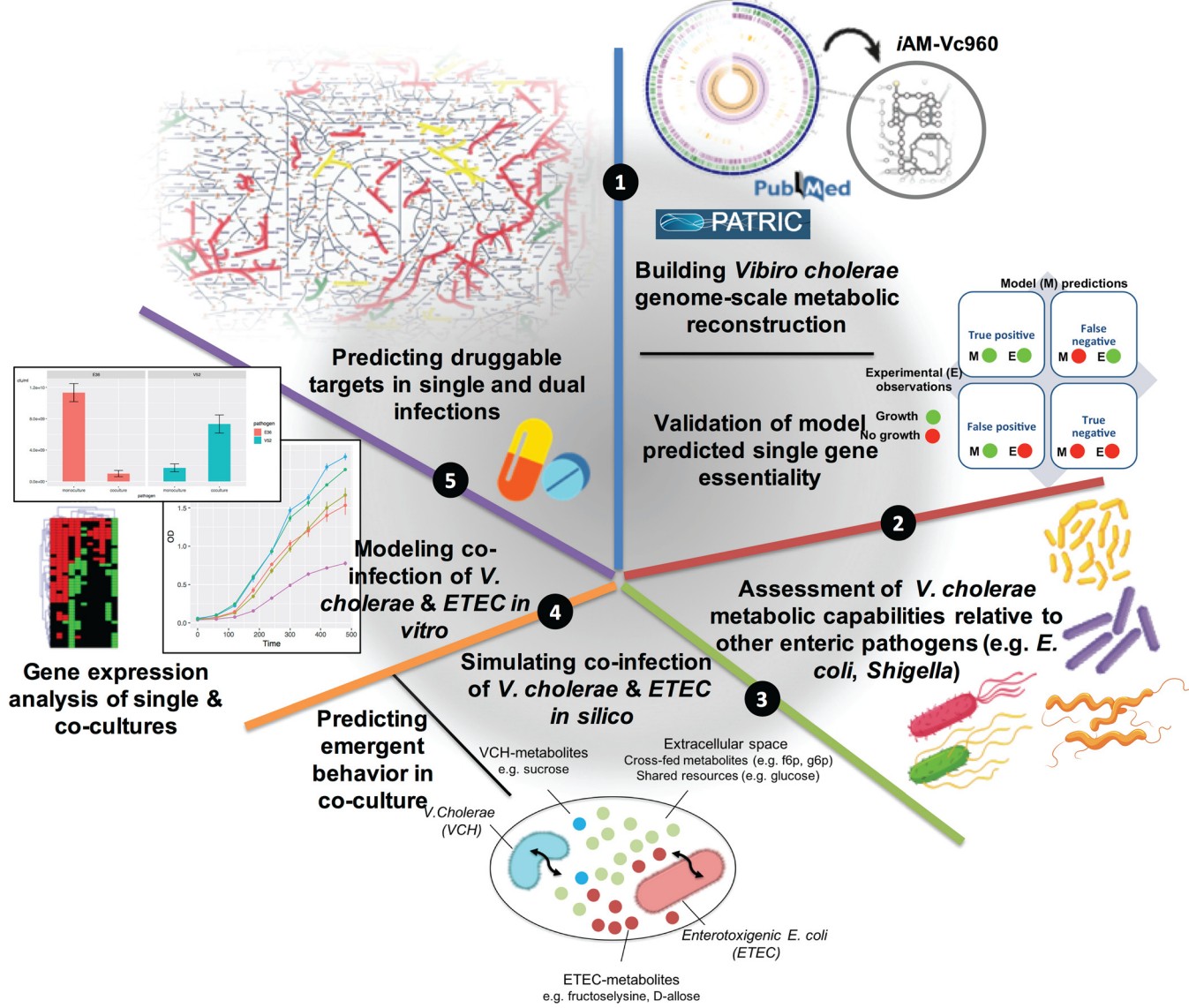

**FIG 1** Overview of the study design.

38 min and 147 min for fast and slow growth, respectively. Hence, the *i*AM-Vc960 predicted doubling time was within the expected range.

In order to further validate *i*AM-Vc960 predictions, we tested if *i*AM-Vc960 could correctly predict gene essentiality. Multiple attempts have been made to generate definitive lists of essential genes, but there are still many discrepancies between these studies even for a model bacterium such as *E. coli* strain K-12 (39). We thus compiled a high-confidence set of genes ($n = 223$; see Table S2 at https://github .com/alyamahmoud/coinfection_modeling/blob/master/supplementary_material/ supplementary_tables.xlsx) that have been shown to be critical for *V. cholerae* growth and survival from three independent previously published studies (40–42). In rich medium (Luria-Bertani broth [LB]), *i*AM-Vc960 correctly predicted 71% of the experi- mentally verified metabolic gene knockouts (see Table S2 at https://github.com/ alyamahmoud/coinfection_modeling/blob/master/supplementary_material/ supplementary_tables.xlsx). In a second step, we also used gene essentiality data for *V. cholerae* strain C6706, a closely related O1 El Tor isolate, obtained from the Online GEne Essentiality (OGEE) database (4, 5), which contains information for essential ($n = 458$)

**A**

| | iAM-Vc960 |
|---|---|
| **Genes** | 960 |
| **Reactions** | 2172 |
| Gene associated (metabolic/transport) | 1570 |
| No gene association (metabolic) | 118 |
| No gene association (transport) | 89 |
| Exchange | 383 |
| Demand/Sink | 11 |
| Biomass | 1 |
| Blocked | 1197 |
| **Metabolites** | |
| Unique metabolites | 1088 |
| Cytoplasmic | 912 |
| Periplasm | 441 |
| Extracellular | 388 |

**B**

Predictions

| | Experimental | | | |
|---|---|---|---|---|
| **iAM-Vc951** | essential | non-essential | Accuracy (p value) | MCC |
| essential | 94 | 67 | 87% (p-value < 2.2e-16) | 0.54 |
| non-essential | 51 | 692 | | |

**FIG 2** *V. cholerae* genome-scale metabolic model *i*AM-Vc960 description and performance evaluation. (A) *V. cholerae* GE statistics (*i*AM-Vc960). (B) Comparison of *i*AM-Vc960 gene essentiality predictions (simulating *in vitro* growth conditions in LB) showed 87% accuracy compared to single gene deletion experiments from OGEE essential (*n* = 458) and nonessential (*n* = 758) gene data sets. *In silico* gene essentiality was graded according to the percentage of reduction in growth rate compared to wild type. The Fisher exact test as well as the Mathew correlation coefficient (MCC) was used to compute significance of overlapping consistent predictions for *i*AM-Vc960. See Table S2 at https://github.com/alyamahmoud/coinfection_modeling/blob/master/supplementary_material/supplementary_tables .xlsx for details.

and nonessential (*n* = 3,144) genes (see supplementary text for a comment on serotype differences at https://github.com/alyamahmoud/coinfection_modeling/blob/master/supplementary_material/supplementary_text.docx). The overall accuracy of *i*AM-Vc960 in reproducing OGEE essentiality (and nonessentiality) data was 87% (Fig. 2B) (see supplementary text at https://github.com/alyamahmoud/coinfection_modeling/blob/master/supplementary_material/supplementary_text.docx for details). Overall, *i*AM-Vc960 predicted 225 and 171 genes to be essential for optimal *V. cholerae* growth in minimal and rich media, respectively.

The agreement between the experimental gene essentiality data, obtained from previously published studies, and the computational results, generated in the current study, in terms of growth and single gene essentiality predictions, on the whole, validates the content of the reconstruction, the modeling procedure, and the objective function definition (Fig. 1, step 1). As such, *i*AM-Vc960 is a high-quality manually curated genome-scale model that can simulate *V. cholerae* metabolism and thus can be used to predict phenotypic behavior of *V. cholerae* in response to different perturbations (e.g., culture conditions, interaction partners, etc.). This prompted us to systematically and comprehensively assess the metabolic capabilities of *V. cholerae* to study how the pathogen adapts its network across the different growth conditions and assess the relative metabolic capacity of *V. cholerae* in relation to other enteric pathogens, as well as how the pathogen's growth capabilities and gene essentiality are impacted in the presence of other coinfecting pathogens.

***V. cholerae* has restricted metabolic capabilities compared to *E. coli* and *Shigella*.** Since enteric bacterial pathogens span several genera including *Escherichia*, *Salmonella*, and *Shigella*, we thought it would be relevant to assess the metabolic capabilities of *V. cholerae* in relation to other pathogens that cause diarrhea (Fig. 1, step 2). Using *i*AM-Vc960, we simulated growth capabilities of *V. cholerae* relative to a set of previously published (13) GEMs of 55 strains of *E. coli* (both commensal and pathogenic) and *Shigella* species on minimal medium with 656 different growth-supporting carbon, nitrogen, phosphorus, and sulfur sources under aerobic and anaerobic conditions (13, 14). *i*AM-Vc960 model size was in line with the smaller genome size of *V. cholerae* than of *E. coli* and *Shigella* (Fig. 3A), where *V. cholerae* has 3,855 open reading frames (ORFs) while *Shigella* and *E. coli* each have on average 4,199 and 4,663 ORFs, respectively. Nevertheless, *i*AM-Vc960 metabolic genes covered 25% of *V. cholerae* ORFs (43). Notably, *i*JO1366, the most well developed and curated genome-scale metabolic model, covers 29% of *E. coli* strain K-12 substrain MG1655 ORFs. On average, *Shigella* and *E. coli* GEMs covered 27% and 29%, respectively, of the corresponding species ORFs.

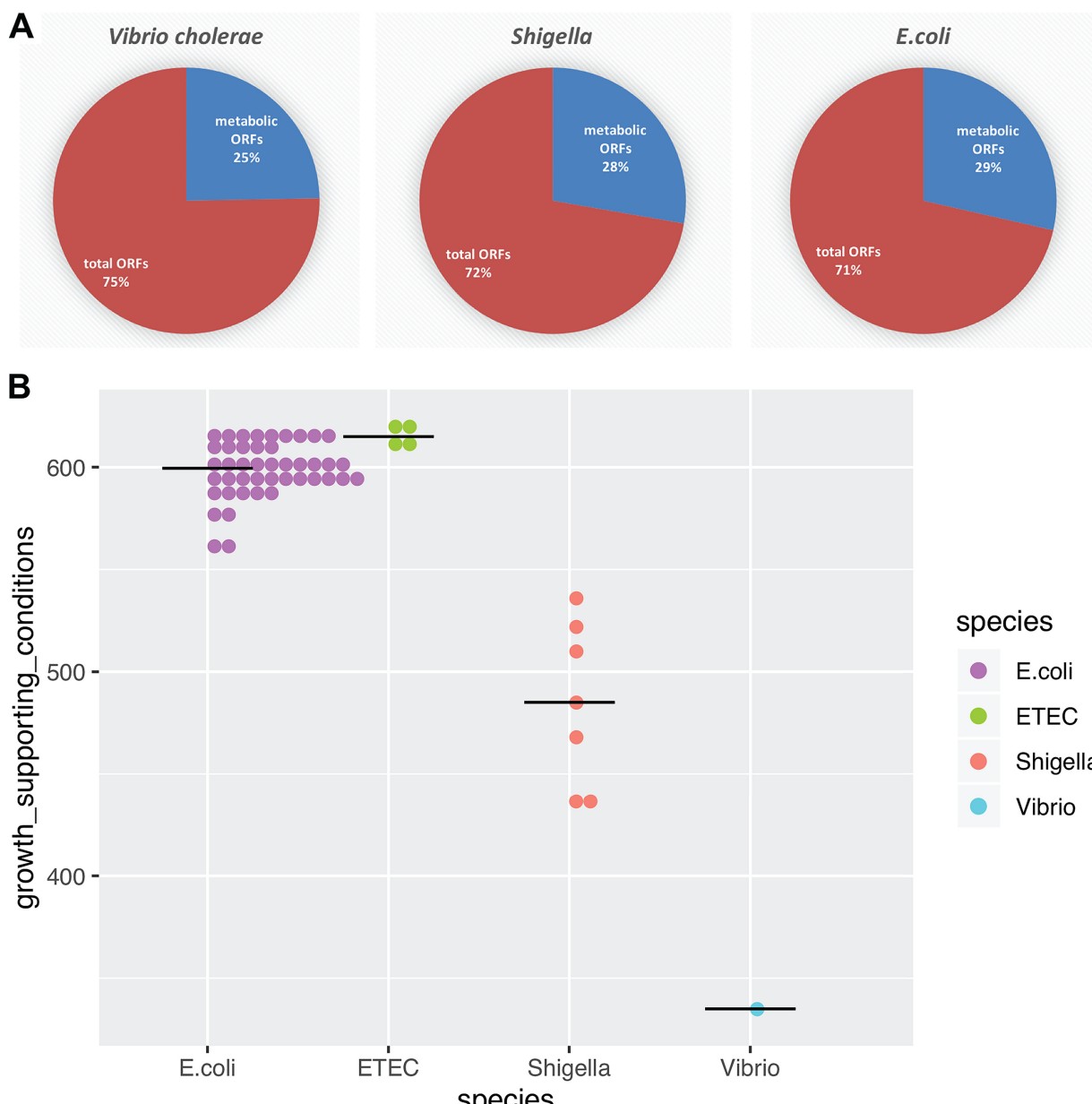

**FIG 3** Functional assessment of *V. cholerae* metabolic capabilities relative to *E. coli* and *Shigella*. (A) Proportion of metabolic genes included as GPR in GEMs of *E. coli* and *Shigella* (13) and *V. cholerae* (this study) relative to total number of ORFs in each species. (B and C) Assessment of *i*AM-Vc960 metabolic capabilities compared to a set of 55 *E. coli* and *Shigella* strains (13) by unique growth-supporting conditions. Predicted metabolic phenotypes under the variable growth-supporting nutrient conditions composed of different carbon, nitrogen, phosphorus, and sulfur nutrient sources under aerobic and anaerobic conditions. Strains were clustered based on their ability to sustain growth in each different environment. Columns in panel B represent individual strains, and rows represent different nutrient conditions. *i*AM-Vc960 coclustered with *Shigella boydii* CDC3083-94, *Shigella boydii* Sb227, and *Shigella dysenteriae* Sd197. Tables S4 and S5 at https://github.com/alyamahmoud/coinfection_modeling/blob/master/supplementary_material/supplementary_tables.xlsx provide all details about the simulation conditions for the alternative nutrient sources. A growth rate of 0.01 was used as the cutoff for binarizing the simulation results and was used to construct the heatmap in panel B.

We first confirmed known metabolic differences for distinguishing *V. cholerae* from other enteric pathogens (Fig. 3B and C). For instance, *i*AM-Vc960 predicted the ability of *V. cholerae* to utilize sucrose as sole carbon source (44, 45). *i*AM-Vc960 could not utilize arginine as sole carbon or nitrogen sources, while all *E. coli* and *Shigella* models were able to utilize arginine under aerobic conditions (46, 47) in line with the frequent usage of the absence of arginine metabolism for characterizing *V. cholerae* (48). Similarly, while *E. coli* and *Shigella* were able to utilize *myo*-inositol as sole phosphorus

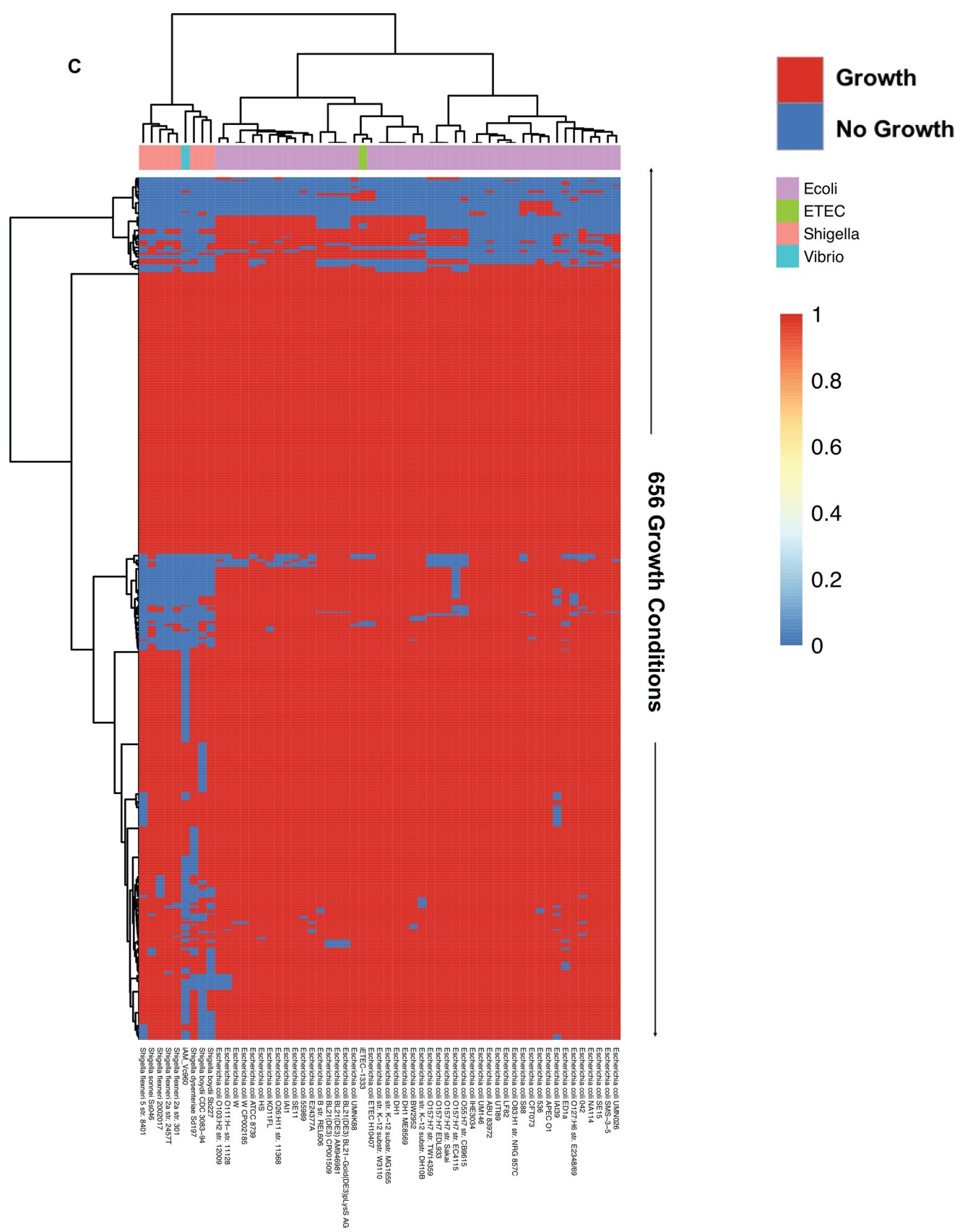

**FIG 3** (Continued)

source, iAM-Vc960 predicted the failure of *V. cholerae* to grow when no other phosphorus source is present in the medium (46). Further, iAM-Vc960 also correctly predicted the ability of *V. cholerae* to utilize trehalose or mannitol as alternative carbon sources under both aerobic and anaerobic conditions (47, 48).

In contrast to *E. coli*, the *V. cholerae* model displayed a large loss of catabolic capabilities across the 656 tested growth conditions (Fig. 3B and C; see Tables S4 and S5 at https://github.com/alyamahmoud/coinfection_modeling/blob/master/ supplementary_material/supplementary_tables.xlsx). This computational result implies that *V. cholerae*, similarly to *Shigella* and several pathogenic *E. coli* strains (49), might have lost catabolic pathways for many nutrient sources. Model predictions showed that *V. cholerae* was able to grow under 51% ($n = 336$) of the simulated growth conditions, while *E. coli* and *Shigella* were able to grow, on average, under 92% ($n = 602$) and 75% ($n = 493$) of the tested growth conditions, respectively (Fig. 3B; see Tables S4 and S5 at https://github.com/alyamahmoud/coinfection_modeling/blob/master/supplementary _material/supplementary_tables.xlsx), implying that *V. cholerae* has less versatile metabolic capabilities than either *E. coli* or *Shigella*. In fact, *V. cholerae* metabolic capabilities were more similar to *Shigella* than to *E. coli* (Fig. 3C). The *V. cholerae* model completely lost the capability to sustain growth on nutrient sources for which most of the *E. coli* and *Shigella* models had growth capabilities. Some of these nutrients include D-lactate, D-fumarate, lactose, L-alanine-glutamate, uridine, xanthosine, thymidine, R-glycerate, sn-glycero-3-phosphoethanolamine, 4-hydroxy-L-threonine, L-asparagine, L-proline, L-arabinose, and L-xylulose as carbon sources as well as nitrate, nitrite (50), ornithine, L-proline, agmatine, uracil, and putrescine (51) as nitrogen sources, and myo-inositol-hexakisphosphate as phosphorus source. Further, most *Shigella* models and iAM-Vc960 were unable to sustain growth on chitobiose, D-malate, D-sorbitol, L-fucose, ethanolamine, galactitol, propionate, D-galactonate, choline, and allantoin as sole carbon sources as well as hypoxanthine, inosine, and urea as nitrogen sources, whereas almost all other *E. coli* models examined were able to sustain growth under the same conditions.

Several tests based on nutrient utilization are routinely used to distinguish between pathogens that cause diarrhea. Using GEMs of enteric pathogens can aid in predicting potential metabolite markers that, upon experimental validation, could be used in clinical practice to diagnose the causative agent of diarrhea or an enteric pathogenesis in general.

**Predicted expanded growth capabilities of *V. cholerae* in coculture with ETEC.** Computational approaches modeling metabolic fluxes between organisms can be used to provide a mechanistic understanding of interaction patterns between different microbes (17, 21, 52, 53). An emergent behavior in coculture will also relate to the extent of overlapping resources between the component species as well as whether or not there will be any cross-fed substrates (22). Using *V. cholerae* as our model organism, we wanted to investigate how the metabolic capabilities (as proxy of growth capabilities) of *V. cholerae* will vary if other coinfecting pathogens are involved (Fig. 1, step 3). We thus set to model coinfections of *V. cholerae* and ETEC. *V. cholerae* (~25%) followed by ETEC (~15%) is the most prevalent bacterial pathogen causing diarrheal diseases in the developing world (30). These bacteria are representative of species found in the same environment and are both involved in enteric pathogenesis. In particular, the choice of these species was inspired by the recurrent dual infections of both species in hospitalized patients due to diarrhea (30–32). The antibody titer against cholera toxin (but not against heat-stable or heat-labile toxins produced by ETEC) was also found to increase in cases of dual infections of *V. cholerae* and ETEC relative to single *V. cholerae* infections (31), although no mechanistic explanation was attributed to these variations. *V. cholerae* V52 was also observed to be virulent against several other Gram-negative species including *E. coli* although ETEC was not tested (54).

To investigate the behavior of the individual pathogens in coinfection relative to their single infections, we used iAM-Vc960 and a previously reconstructed GEM of ETEC, iETEC1333 (13), to simulate the growth of *V. cholerae* and ETEC in a single shared

environment (55, 56). Metabolic genes, metabolic reactions, and metabolites were compared across the species-specific networks. *i*AM-Vc960 and *i*ETEC-1333 had 1,672 metabolites in common. This represented 96% and 85% of *V. cholerae* and ETEC total metabolites, respectively. To distinguish between shared and species-specific metabolites, each organism was represented as a separate compartment (Fig. 4A) with a shared space representing the coculture/infection medium. Twenty-three percent ($n = 380$) of the common metabolites between the two models were amenable to exchange by being available in the shared extracellular space (Fig. 4A). In total, the coculture model, *i*Co-Culture2993, had 4,550 reactions, 3,335 metabolites, and 2,293 genes. The objective function was set to maximize the biomass function of each pathogen, simulating growth of both species at 1:1 composition (see Materials and Methods and see also supplementary text at https://github.com/alyamahmoud/coinfection_modeling/blob/master/supplementary_material/supplementary_text.docx for details in development and refinement of the coculture model).

We then used the same set of 656 growth conditions to assess the difference in metabolic capabilities of *V. cholerae* and ETEC in single infections and coinfections. All three models (*i*AM-Vc960, *i*ETEC1333, and *i*Co-Culture2993) were able to grow under 51% ($n = 333$) of the tested growth conditions. ETEC was able to grow under 42% ($n = 277$) of the growth conditions that *V. cholerae* was unable to utilize in single culture. However, *i*Co-Culture2993 acquired the capability to grow under the same conditions (Fig. 4B; see also Tables S4 and S5 at https://github.com/alyamahmoud/coinfection_modeling/blob/master/supplementary_material/supplementary_tables.xlsx). A closer look revealed that most of those acquired capabilities were due to ample cross-feeding opportunities enabled by the ETEC model. For instance, *i*AM-Vc960 is unable to grow on putrescine as sole nitrogen or carbon source. *i*ETEC1333 and *i*Co-culture2993, however, are able to degrade putrescine into glutamate by putrescine transaminase (*patA*: ETEC_3343) or into glutamate and succinate through the gamma-glutamyl putrescine synthetase (*puuA*: ETEC_1401)/oxidoreductase (*puuB*: ETEC_1405) pathway, both being absent in the *V. cholerae* genome. Similarly, *V. cholerae* cannot catabolize uridine (and xanthine) whereas ETEC can degrade uridine, xanthine, and xanthosine into ribose as it possesses pyrimidine-specific ribonucleoside hydrolases (*RihA*, *RihB*, and *RihC*: ETEC_0680, ETEC_2297, and ETEC_0030) which can potentially be cross-fed to *V. cholerae*. In addition, several D-amino acids were observed to be cross-fed where they are degraded by ETEC into forms that can be utilized by *V. cholerae*, e.g., D-allose which is degraded by ETEC D-allose kinase (*alsK*: ETEC_4394) into fructose-6-phosphate that can be cross-fed to *V. cholerae*. Similarly, fructoselysine is metabolized by ETEC fructoselysine kinase (*frlD*: ETEC_3624) and fructoselysine 6-phosphate deglycase (*frlB*: ETEC_3622) into glucose-6-phosphate which can be cross-fed to *V. cholerae*. None of those genes have been identified in the genome of *V. cholerae* to date (determined via searching the annotated genome of *V. cholerae* O1 biovar El Tor strain N16961 in PATRIC [57] and UniProt [58] and the annotated genome of *V. cholerae* V52 generated in this study as well as two other assemblies, GCF_001857545.1 and GCF_000167935.2, retrieved through PATRIC [57]).

Overall, *i*ETEC1333, *i*AM-Vc960, and *i*Co-Culture2993 were able to grow in 94% ($n = 614$), 51% ($n = 336$), and 93% ($n = 613$) of the simulated growth conditions, respectively (Fig. 4B). As such, we predict that *V. cholerae* metabolic capabilities are expanded in coinfections with ETEC relative to *V. cholerae* single infections while ETEC metabolic capabilities are almost not affected where the main differences between the two species lie in their capability to take up and catabolize various nutrient sources. Our modeling approach thus provides mechanistic insights into the observed increase in cholera infection severity in clinical patients who demonstrated increased antibody titers against cholera (and not ETEC) toxin in case of coinfections by the two enteric pathogens (31).

**A**

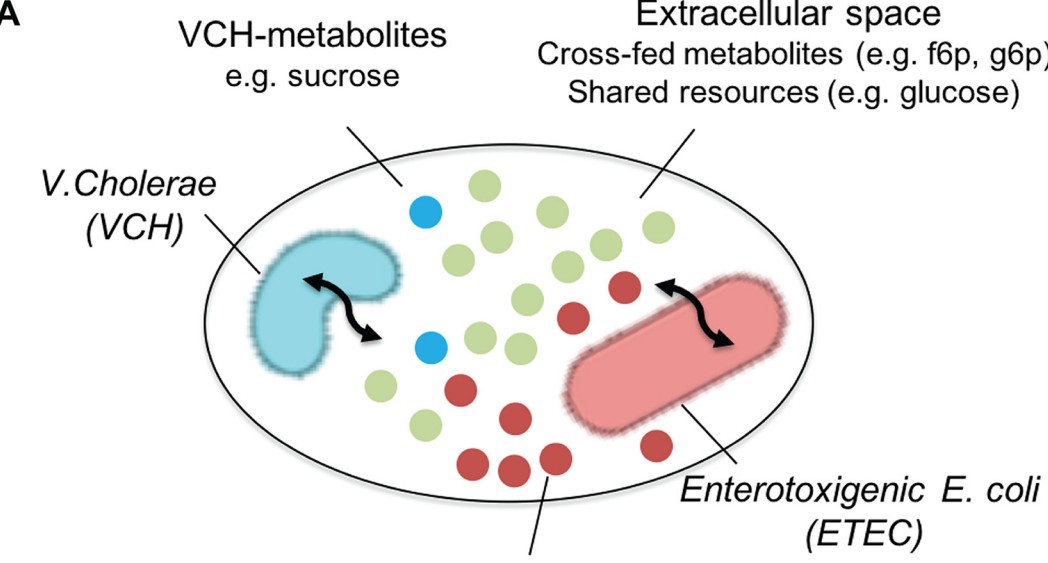

**B**

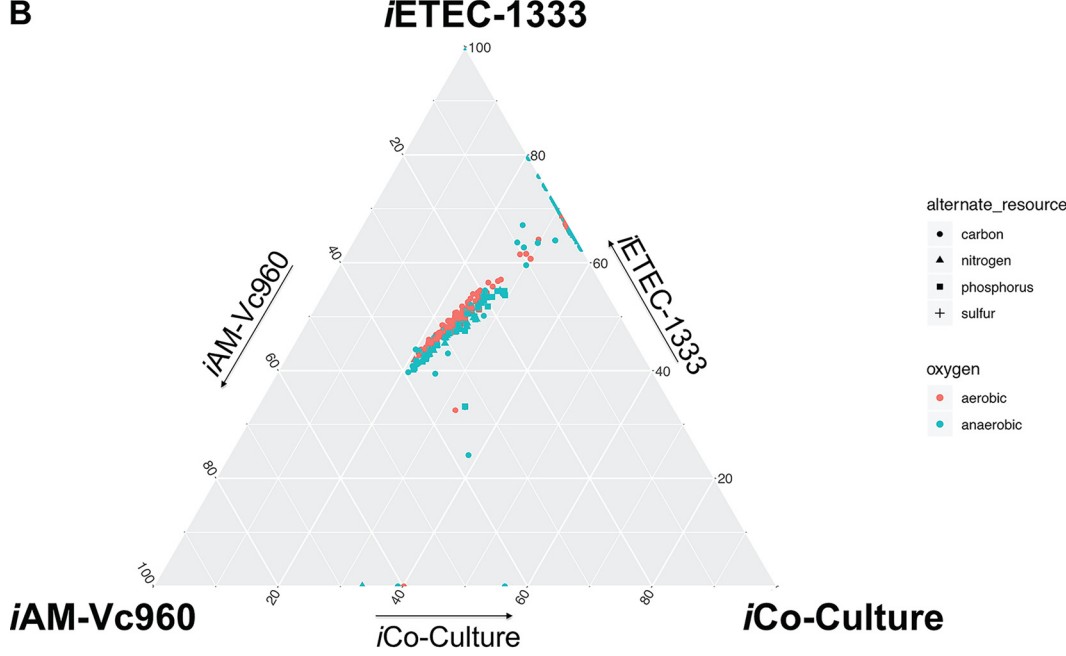

**10 h**

**FIG 4** Computational modeling and *in vitro* coculture of *V. cholerae* and ETEC coinfection. (A) Schematic showing the modeling framework used to simulate growth of *V. cholerae* and ETEC in a shared environment. (B) Ternary plot showing 600+ growth conditions to compare the metabolic capabilities of *V. cholerae* and ETEC monocultures relative to their coculture. Values used for plotting are flux rates in the biomass objective function of each model and are meant to show the ability to grow or not grow under the respective growth condition rather than the flux value. No change in the overall plot was observed when using normalized values relative to standard growth conditions (aerobic conditions + glucose/ammonia/phosphate/sulfate). (C) Quantification of *V. cholerae* and ETEC CFU in monocultures and cocultures over 10 h for the CFU (pooled technical replicates of $n = 3$ biological replicates) in M9 minimal medium supplemented with 0.5% glucose, 1 mM MgSO$_4$, and 0.1 mM CaCl$_2$, with 5 $\mu$l spotted at each time point. Data shown as mean $\pm$ SD for three biological replicates. (D) Dynamics of *V. cholerae* in coculture with enterotoxigenic *E. coli* and in monoculture. Data shown as mean $\pm$ SD for three biological replicates.

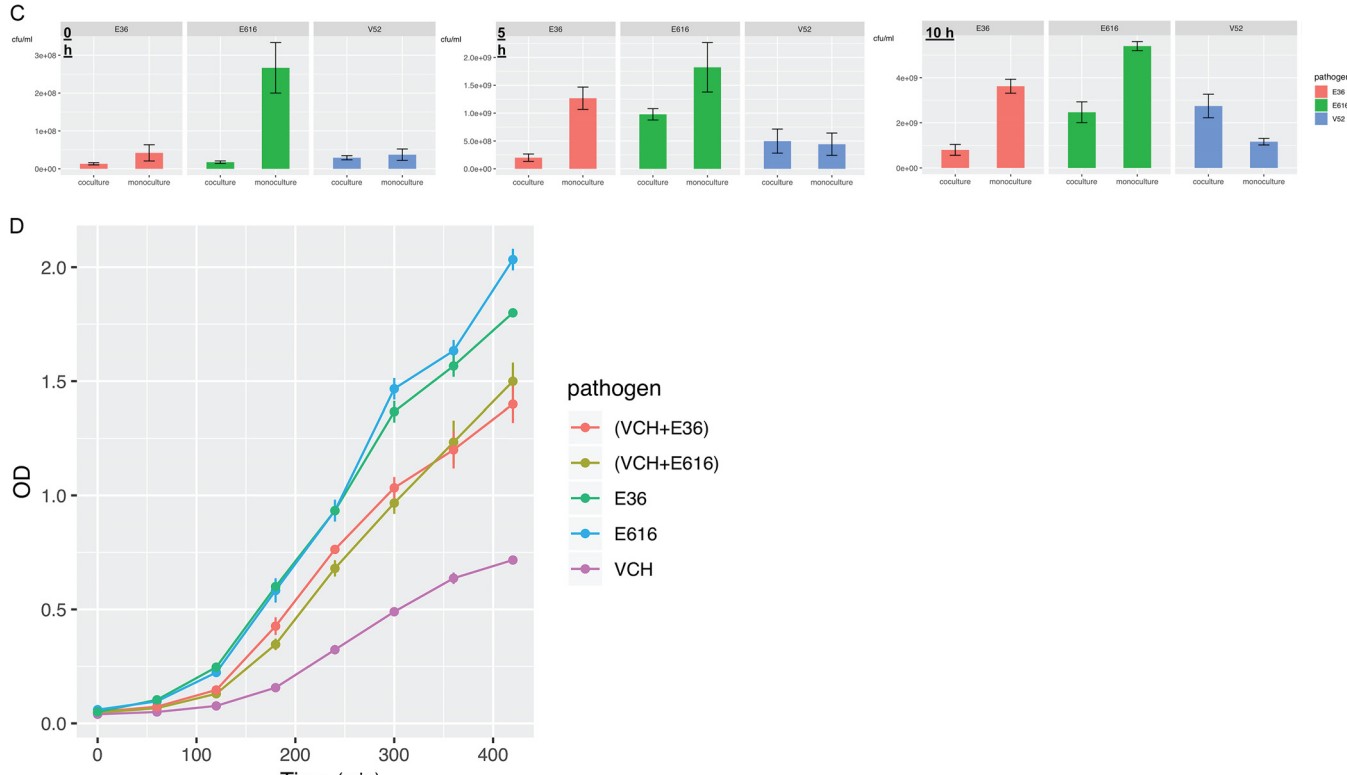

**FIG 4** (Continued)

**Growth of *V. cholerae* is enhanced when cocultured with ETEC *in vitro*.** To validate our predictions, we employed single- and coculture *in vitro* experiments (Fig. 1, step 4) to assess the predictions made by our enteric pathogen coinfection model (Fig. 4C and D and see Table S6 at https://github.com/alyamahmoud/coinfection_modeling/blob/master/supplementary_material/supplementary_tables.xlsx). To this end, we developed a robust *in vitro* coculture system of *V. cholerae* V52 and two different ETEC strains (E36 and E616) in M9 minimal medium supplemented with glucose (Fig. 4C and D). All three tested strains (V52, E36, and E616) are clinical isolates that have been sequenced and characterized previously (59, 60) (see supplementary text at https://github.com/alyamahmoud/coinfection_modeling/blob/master/supplementary_material/supplementary_text.docx for details on strain selection and sequencing performed as part of the current study). We determined the impact of the coculture on each strain's growth by comparing single-culture abundance over 10 h of growth to the abundance of each strain in coculture at the same time (determined using CFU counting; all strains were in transition or stationary phase). E36 and E616 were shown to have diminished ability to grow in coculture with *V. cholerae* V52. In contrast, growth of *V. cholerae* V52 was strongly enhanced under coculture conditions (Fig. 4C and D).

The growth data regarding *V. cholerae* V52 were in agreement with the modeling predictions. When comparing maximal abundances, cross-feeding and competitive interactions were already apparent. *V. cholerae* V52 reached higher maximal bacterial counts in *V. cholerae* V52/ETEC E36 (unpaired two-sided Wilcoxon: shift 5.8e+09, 90% confidence interval 3.8e+09 to 6.8e+09, *P* value 0.07) and in *V. cholerae* V52/ETEC E616 (unpaired two-sided Wilcoxon: shift 5.6e+09, 90% confidence interval 4.4e+09 to 8.8e+09, *P* value 0.1) cocultures (Fig. 4C and D). The maximum cell number of both ETEC strains tended to be lower when competing with *V. cholerae* V52 than when grown alone (unpaired two-sided Wilcoxon E36: shift −1.06e+10, 90% confidence interval −1.14e+10 to −8.60e+09, *P* value 0.07; unpaired two-sided Wilcoxon E616:

shift −6e+09, 90% confidence interval −9.4e+09 to −2.0e+09, P value 0.1). Finally, according to maximal bacterial counts, E36 was more negatively affected by the presence of *V. cholerae* V52 than E616 (unpaired two-sided Wilcoxon E36: shift −6.4e+09, 90% confidence interval −9.4e+09 to −5.2e+09, P value 0.1).

Although our modeling procedure predicted and explained the increase in *V. cholerae* growth capabilities when cocultured with ETEC, the decrease in abundance of ETEC in *V. cholerae* V52/ETEC cocultures was not captured by our metabolic models. *V. cholerae* V52 was previously found to be highly virulent against several Gram-negative bacteria, including *E. coli* and *Salmonella enterica* serovar Typhimurium, due to the type VI secretion system (T6SS) (54). Although ETEC was not tested for in these experiments, it is expected that ETEC would behave similarly to closely related pathogenic *E. coli* strains (enteropathogenic *E. coli* [EPEC] and enterohemorrhagic *E. coli* [EHEC]). Thus, the decrease in ETEC growth is very likely mediated by nonmetabolic factors. We also focus on the improved growth of *V. cholerae* since this is of potential clinical relevance and since the decrease in ETEC growth in *V. cholerae* cocultures has been investigated before.

**Altered gene expression in single- and multispecies cocultures.** To assess the level of genetic perturbations due to addition of ETEC as an interaction partner to *V. cholerae* cultures, we conducted a dual RNAseq analysis (61–64) of *V. cholerae* cocultures (Fig. 1, step 4) with each of the two ETEC strains (E36 and E616). We then compared the gene expression levels for each pathogen to its single culture (see Materials and Methods and see also Tables S7 to S10 at https://github.com/alyamahmoud/coinfection _modeling/blob/master/supplementary_material/supplementary_tables.xlsx). Through principal-component analysis (PCA) (Fig. 5; see also Fig. S5 at https://github.com/ alyamahmoud/coinfection_modeling/blob/master/supplementary_material/ supplementary_text.docx), we found that the coculture expression data clustered independently from single-culture data, indicating that the transcriptome of *V. cholerae* is distinct during coculture compared to single culture. The expression of 20% of the *V. cholerae* quantifiable transcriptome was significantly altered when either strain of ETEC was added to the culture. In particular, 15 to 17% of *V. cholerae* genome was upregulated while 4 to 5% was downregulated in *V. cholerae* coculture with ETEC relative to its single culture. *V. cholerae* differentially expressed genes were enriched in diverse metabolic processes spanning amino acid metabolism like tyrosine and L-phenylalanine (P value <0.01, odds ratio >10) as well as carbohydrate metabolic processes (P value <0.05, odds ratio = 2.630409) (Fig. 5; also see Tables S9 and S10 at https://github .com/alyamahmoud/coinfection_modeling/blob/master/supplementary_material/ supplementary_tables.xlsx). Upregulation of certain amino acid biosynthesis pathways, which can be catabolized by both species, highlights that, despite potential cross-feeding between the two pathogens, the presence of more than one infectious agent might eventually lead to competition (65). Further, in support of non-metabolism-mediated suppression in growth observed for ETEC, E36 differentially expressed processes were significantly enriched in taxis and chemotaxis GO terms (P value = 3.8e−05 and odds ratio >20). Also, in line with previous reports (54, 60) about T6SS expression levels, T6SS components were constitutively expressed in *V. cholerae* V52 in both single culture and cocultures (see Tables S9 and S10 at https://github.com/alyamahmoud/ coinfection_modeling/blob/master/supplementary_material/supplementary_tables .xlsx).

In line with predicted cross-feeding interactions between *V. cholerae* and ETEC, we found that gamma-glutamyl putrescine oxidase (*puuB*) and putrescine utilization regulator (*puuR*) as well as several putrescine transporters were indeed significantly upregulated in E616/V52 coculture relative to E616 single culture (logFC [fold change] >1.5, adjusted P value <0.05). Furthermore, neither *patA* nor *puuB* was expressed in *V. cholerae* V52. Similarly, ribose-5-phosphate isomerase B (*rpiB*) and transcriptional regulator of D-allose utilization (*rpiR*) were significantly upregulated in E616/V52 coculture relative to E616 single culture (logFC >2, adjusted P value <0.005)

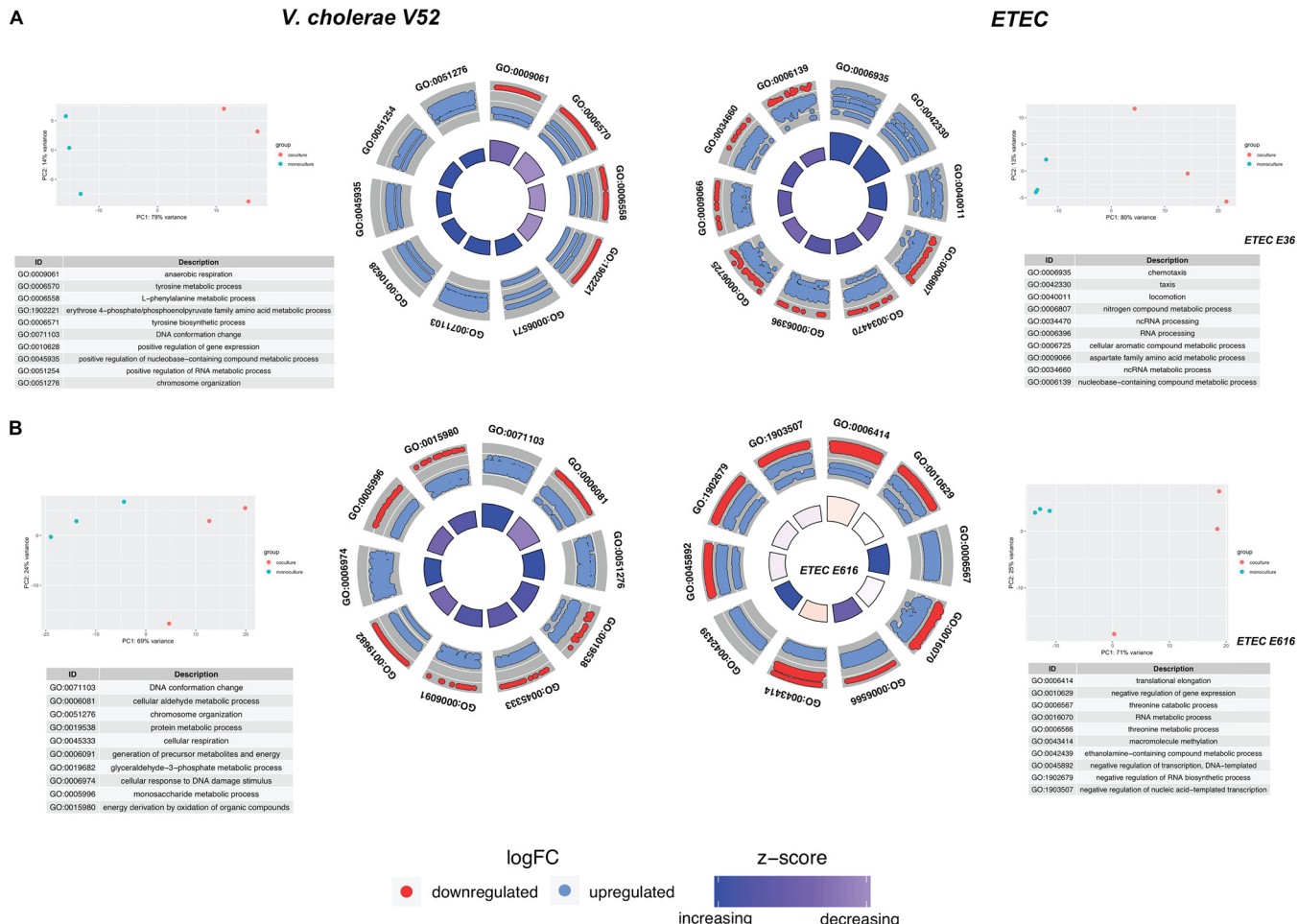

**FIG 5** Dual RNAseq analysis of *V. cholerae* and ETEC in coculture. GO enrichment of *V. cholerae* differentially expressed FIGfams in coculture with ETEC E36 (A) and E616 (B) relative to its single culture. Up- and downregulation are in *V. cholerae* when in coculture relative to its monoculture. Z-score is calculated according to GOplot (up-down/√*count*) where *up* and *down* are the number of assigned genes upregulated (logFC > 0) in the data or downregulated (logFC < 0), respectively. PCA plots show that the monocultures are clustering differently from the cocultures for either species.

and were not expressed in *V. cholerae* V52. Lastly, transcriptional regulator of fructose-lysine utilization operon (*frlR*), fructoselysine 6-kinase (*frlD*), fructoselysine 3-epimerase (*frlC*), and fructoselysine-6-phosphate deglycase (*frlB*) were also significantly upregulated in E616/V52 coculture relative to E616 single culture (logFC >1 to 1.5, adjusted *P* value <0.05).

Interestingly, expression levels of bacteriocins' related genes in ETEC strains showed that colicins' production and tolerance genes were significantly upregulated in E616 coculture with *V. cholerae* V52 relative to the individually grown E616 (see Table S8 at https://github.com/alyamahmoud/coinfection_modeling/blob/master/supplementary _material/supplementary_tables.xlsx). In contrast, E36, whose growth is more sensitive to cogrowth with *V. cholerae* V52, failed to upregulate genes encoding colicin V production and tolerance genes (see Table S7 at https://github.com/alyamahmoud/ coinfection_modeling/blob/master/supplementary_material/supplementary_tables .xlsx). Colicin V is a peptide antibiotic that members of *Enterobacteriaceae* commonly use to kill closely related bacteria in an attempt to reduce competition for essential nutrients (66). To sum up, the difference in expression levels of genes encoding colicin production and resistance explains why E36 growth was more severely affected when cocultured with *V. cholerae* V52 than with E616 (see Fig. S5 at https://github.com/alyamahmoud/coinfection _modeling/blob/master/supplementary_material/supplementary_text.docx).

RNAseq thus confirmed that there is an emergent behavior in the cocultures and that the observed changes were not due just to variations in inoculum composition or

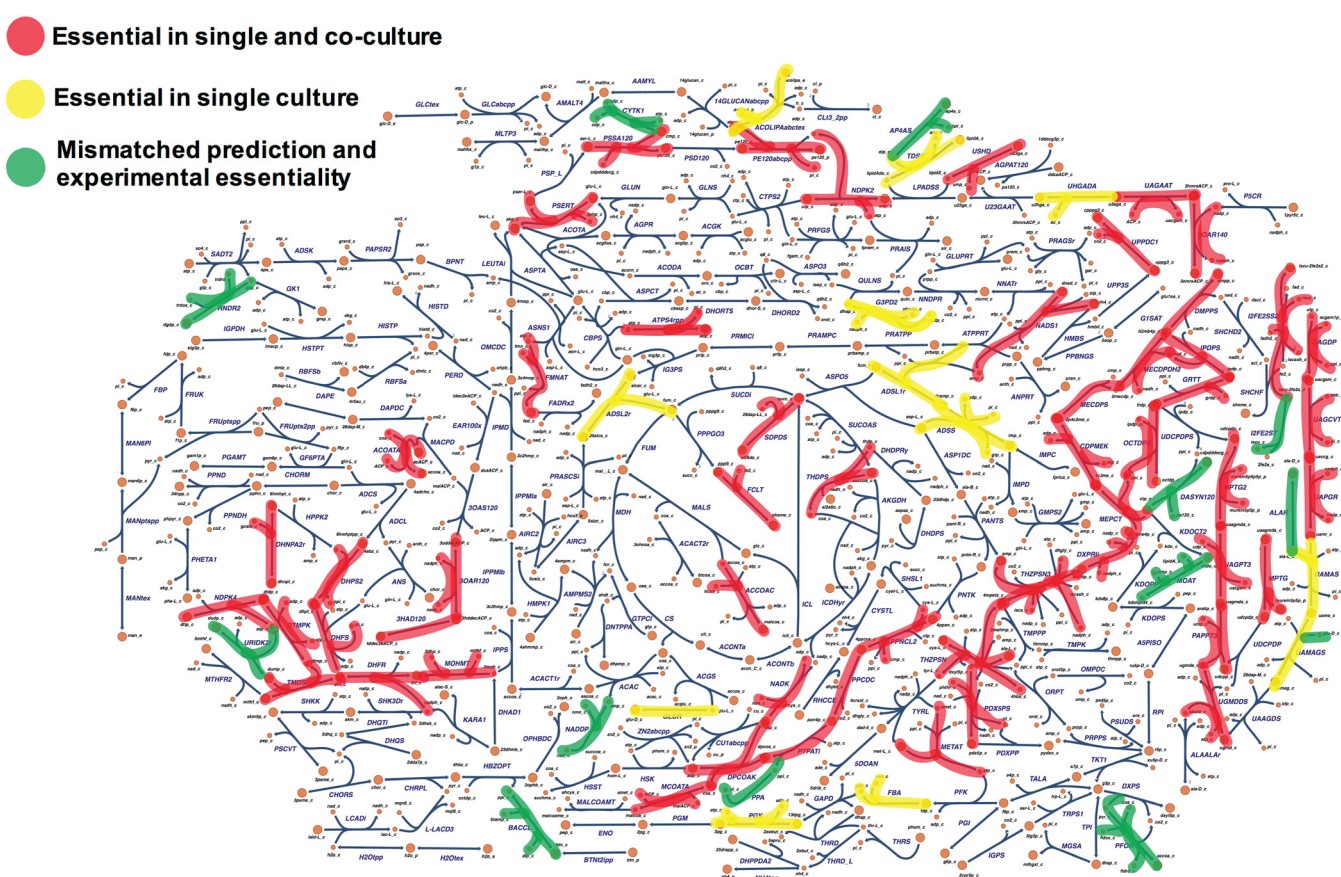

**FIG 6** Comprehensive map of *V. cholerae* essential metabolic genome constructed by projecting the list of experimentally validated essential genes onto our single and coinfection models' predictions. Inhibitors against red targets are expected to have a broader spectrum since they are essential for *V. cholerae* in both single and coinfection scenarios. Inhibitors against yellow targets are essential for *V. cholerae* growth in single-infection scenarios, losing their essentiality only in the presence of other coinfecting species. Green targets indicate a mismatch between model-predicted and experimentally validated essentiality.

the lag phase (64). Taken together, our integrated modeling, coculturing, and transcriptomics approach provided mechanistic insights into the observed increase in cholera infection severity in dual infections with ETEC where ETEC coinfection results in an increased growth of *V. cholerae* due to expanded metabolic capabilities enabled by ETEC. In parallel, *V. cholerae* suppresses ETEC growth by nonmetabolic factors, resulting in an increase in cholera infection severity but not ETEC as monitored by antibody titer against species-specific toxins (31).

**Evaluation of experimentally validated essential genes across single-infection and coinfection models of *V. cholerae*.** The essential genome of a large class of bacterial species has been characterized as it encodes potential targets for antibacterial drug development (39, 67). Interestingly, metabolic genes have predominated in studies of essential genomes of microbial pathogens (67, 68). With this in mind, we attempted to construct a comprehensive map of *V. cholerae* essential metabolic genome (Fig. 1, step 5) by projecting the list of experimentally validated essential genes onto our single-infection and coinfection models' predictions (Fig. 6; see also Table S2 at https://github.com/alyamahmoud/coinfection_modeling/blob/master/supplementary_material/supplementary_tables.xlsx). Selecting targets that are critical in both single-infection and coinfection settings would promote the discovery of novel targets or new combinations of existing antibacterials that would be effective in a broader spectrum of cholera infections. The color scheme of highlighted reactions (Fig. 6) denotes model prediction classification across single infections and coinfections. The red group in Fig. 6 highlights reactions predicted to be sensitive in both single infections and coinfections; this is of particular importance since the efficacy of some of the commonly used treatment drugs

might significantly be altered in the presence of more than one infecting agent. There are several gene deletions associated with reactions for which drugs have not been developed (see Table S2 at https://github.com/alyamahmoud/coinfection_modeling/ blob/master/supplementary_material/supplementary_tables.xlsx). These highlight potential targets for new drug development that may aid in treating enteric pathogenesis. We also note that the green group identifies reactions that were missed by the models and highlights areas for future model refinement.

Out of the 80 metabolic genes that have been experimentally shown to be essential for *V. cholerae* growth and survival across several studies (see Table S2 at https://github .com/alyamahmoud/coinfection_modeling/blob/master/supplementary_material/ supplementary_tables.xlsx), our coculture model predicted 47 genes to be critical for *V. cholerae* growth even when a more metabolically versatile enteropathogen like ETEC is added to the culture irrespective of the variation in species composition (see Materials and Methods and see also Table S2 at https://github.com/alyamahmoud/coinfection _modeling/blob/master/supplementary_material/supplementary_tables.xlsx for details). This set of 47 genes (Fig. 6, red) represents potential drug targets that are predicted to be effective in killing *V. cholerae* whether it is the sole cause of diarrhea or as part of a polymicrobial infection. Most of these enzymes were involved in cofactor biosynthesis (e.g., coenzyme A [CoA], tetrahydrofolate, flavin adenine dinucleotide [FAD], pyridoxone-5-phosphate, pantothenate, and iron-sulfur cluster) and isoprenoid and porphyrin metabolism as well as pyrimidine metabolism (Fig. 6). Inhibitors of several of those enzymes have already been reported to have a bactericidal effect (57) in *V. cholerae* as well as in other enteric and nonenteric pathogens (see Table S2 at https://github.com/alyamahmoud/coinfection_modeling/blob/master/supplementary _material/supplementary_tables.xlsx). For instance, phosphopantetheine adenylyl-transferase and thymidylate synthase have been already reported as drug targets (57) in *V. cholerae* and ETEC E616. *N*-Acetylglucosamine transferase is a promising drug target for *Salmonella enterica* while dephospho-CoA kinase has been shown to be an interesting drug target in ETEC E616 and *Shigella flexneri* (57). Interestingly, 12 *V. cholerae* genes were also predicted to totally lose their essentiality in dual infections with ETEC. Some of those were involved in *de novo* purine metabolism (VC1126: *purB*, VC2602: *purA*) and carbohydrate degradation (VC0477: *pgk*, VC0478: *fbaA*), implying that *V. cholerae* is probably depending on ETEC to salvage these nutrients.

ATP synthase subunits were essential for *V. cholerae* growth in single cultures as predicted by *i*AM-Vc960. Deletion of any of the 7 genes of the F0/F1 ATP synthase locus in the coculture model resulted in a species-composition-dependent reduction in reduced optimal growth (see Table S2 at https://github.com/alyamahmoud/coinfection _modeling/blob/master/supplementary_material/supplementary_tables.xlsx). In models simulating high *V. cholerae* abundance relative to ETEC, ATP synthase subunits were essential for optimal coculture growth. In contrast, models simulating higher ETEC abundance relative to *V. cholerae* were less affected when ATP synthase subunits were deleted. F0/F1 ATP synthase genes have been shown to be essential in a variety of bacteria (40, 69–72) and have been recently reported as essential in *V. cholerae* (40). In *E. coli*, however, ATP synthase is not essential (40, 73, 74). Thus, drug inhibitors (acting on ATP synthase subunits) that would normally kill *V. cholerae* in single infections would have decreased efficacy in cases of dual infections with *E. coli*. This suggests that comparison of essential genes between organisms can uncover distinct ecological and physiological requirements for each species (40) and should inspire future experiments to validate our computational predictions. Similarly, sodium-dependent NADH dehydrogenase (Na$^+$-NQR), a key component of the respiratory chain of diverse bacterial species, including pathogenic bacteria, and succinate dehydrogenase subunits were also predicted to lose essentiality for *V. cholerae* growth when cocultured with *E. coli*. Taken together, our *in silico* predictions of variations in essentiality between single-culture and coculture settings highlight the importance of considering both scenarios when prioritizing druggable targets for downstream validation.

## DISCUSSION

Using integrated metabolic modeling, *in vitro* culturing and transcriptomics, we investigated the growth phenotypes and single gene essentiality variations of a representative human pathogen, *V. cholerae*, when implicated in single infections or coinfections. We found that *V. cholerae* growth is enhanced in coinfection scenarios with ETEC. Our modeling procedures explained this increase in *V. cholerae* growth by an expansion in its metabolic capabilities through cross-fed metabolites enabled by ETEC, reproducing observed behavior in patients with dual infections by the two enteric pathogens. We further predicted a core set of essential genes that are critical for *V. cholerae* growth whether it is implicated in single or dual infections with ETEC.

Our modeling approach allowed us to chart possible metabolites that can be cross-fed to *V. cholerae* through ETEC (see Table S5 at https://github.com/alyamahmoud/coinfection_modeling/blob/master/supplementary_material/supplementary_tables.xlsx). Cross-feeding, in which one species produces metabolites consumed by another, has been shown more than often to be adopted by coexisting species across diverse environments (17, 18, 53, 75). Questions like whether the release of cross-fed metabolites or by-products would enhance or enable the growth of other species or whether it will be costless or associated with reduced fitness of the producer are not usually clear. Such questions become of even greater importance when it comes to pathogens since this will have direct impact on the dosage and spectrum of antibiotics used. Our integrative approach provides insights into how to arrive at primary answers to similar questions that should direct future experimental work.

A large fraction of the *V. cholerae* essential genome (36%) (see supplementary text at https://github.com/alyamahmoud/coinfection_modeling/blob/master/supplementary_material/supplementary_text.docx for details) consists of metabolic functions spanning several processes including cell wall biosynthesis, lipid metabolism, and cofactor biosynthesis (76–78). Most essential genes for *V. cholerae* growth whether it was causing single or coinfections were also involved in cofactor biosynthesis. Interestingly, cofactor-use-efficient pathways were often favored by organisms that depend on simple carbon sources under anaerobic conditions (79) resembling growth conditions in the intestine (50, 80), where *V. cholerae* and ETEC establish their infection. The application of this work is of immediate relevance for the choice of antibiotics used in cases of single or polymicrobial infections. Strategies that depend on an increase in dosage of one drug or combining drugs of known efficacy against individual species might not necessarily work when two or more pathogens are operating together. Our findings indicate that the essential transcriptome of *V. cholerae* is distinct during coinfection compared to single infection and highlight the importance of studying pathogen gene essentiality in polymicrobial infections. While replacement fluids are the main treatment line for *V. cholerae* infections, antibiotics are frequently used to lessen the diarrheal purging, decrease the need for rehydration fluids, and shorten the recovery time (23). For other human pathogens, however, antibiotics are the mainstay, and we envision that our framework can be applied to other pathogens and their most frequently reported coinfecting partners. We believe that such an integrative approach could be routinely integrated as part of drug target development pipelines.

An integral part of constraint-based modeling relies on reconciling differences that arise between modeling and experiments (10–12, 81). In our case, coinfection models' simulations predicted an increase in *V. cholerae* growth rate coupled with almost no impact on ETEC growth capabilities. This is in line with recent studies showing that most organisms secrete a broad distribution of metabolically useful compounds without cost under a variety of environmental conditions (53). However, our *in vitro* coculture experiments revealed a significant decrease in ETEC growth rate leading us to conclude, in light of existing literature (54), that the suppression in ETEC growth is potentially mediated by nonmetabolic factors that are not captured by our GEMs.

Although our approach is based on computational predictions and *in vitro* experi-

ments which definitely do not fully recapitulate *in vivo* conditions, our growth phenotype, predicted by coculture models and *in vitro* cocultures, matched observed behavior in patients presenting with diarrhea while being coinfected with both *V. cholerae* and ETEC showing higher antibody titers against cholera toxin relative to patients infected with *V. cholerae* only (31). Nevertheless, we realize that there are other processes that are not accounted for even after integrating data from various sources within the current approach. For instance, the fact that our metabolic model could not predict the decrease in ETEC growth rate implies that this effect is probably mediated by a nonmetabolic factor that is not captured by the metabolic models as such. Future models, building upon the present reconstruction, can expand the modeling scope to account for synthesis and secretion of *V. cholerae* virulence factors (6–9) in an attempt to investigate how the metabolic network of *V. cholerae* impacts the synthesis of its virulence factors. Coculture experiments create an artificial community in a controlled environment and thus provide ideal conditions to test ecological concepts concerning community stability and dynamics that cannot easily be measured in macroecological complex systems (82). However, most parts of the human intestine are hypoxic, vary in pH level (50, 80), and are inhabited by diverse sets of commensal microbes which are not accounted for when solely depending on *in vitro* experiments. Current predictions and experiments thus do not capture several of these factors including temperature, pH changes, signaling, gene regulation, serotype differences, and coexisting commensal microbes (which may account for the absence of the *V. cholerae* growth phenotype when using solid agar or spent medium for coinfection modeling; see Fig. S5 and S6 and supplementary text at https://github.com/alyamahmoud/coinfection_modeling/blob/master/supplementary_material/supplementary_text.docx for details).

Our study investigates a synthetic enteric pathogens community with a combination of *in vitro* single cultures and cocultures, mechanistic modeling, and gene expression analysis. Constraint-based modeling approaches, which can take emergent metabolism into account (34), require high-quality metabolic reconstructions for each community member, which take months of curation effort to obtain (83). However, the modular nature of the modeling approach followed here implies that such approaches can be scaled up to simulate polymicrobial infections as well as coexisting commensal microbes to further prioritize druggable targets that would be effective under an even broader range of infection conditions and complex ecosystems. Collectively, this work illustrates the importance of harnessing the power of integrative predictive modeling coupled with coculture experiments to recognize potential amplification in a pathogen's growth capabilities *a priori* which could contribute to downstream therapeutic and management options.

## MATERIALS AND METHODS

The methods employed for the reconstruction, simulation, and analyses presented in this work are briefly summarized below, with further details regarding the procedures, protocols, calculations, and quality control measures provided in the supplementary material on GitHub. All supplementary tables are available as part of a GitHub repository at https://github.com/alyamahmoud/coinfection_modeling.

**Growth assays and CFU measurements.** Bacterial strains were grown in M9 (Sigma-Aldrich) minimal medium supplemented with 0.5% glucose, 1 mM magnesium sulfate, and 0.1 mM calcium chloride, unless otherwise specified. *V. cholerae* V52 (O37 serogroup) and the enterotoxigenic *Escherichia coli* strains (ETEC E616 and ETEC E36) were a kind gift from Sun Nyunt Wai, Umeå University, Sweden. *V. cholerae* and ETEC were grown either individually (monocultures of V52, E616, and E36) or in combination (cocultures of V52/E616 and V52/E36) at 37°C at 200 rpm. Cocultures were started with equal concentrations of each strain. The absorbance (optical density at 600 nm [$OD_{600}$]) was measured every 1 h over a period of 7 h for the growth curve measurements. Simultaneously, at every hour, an aliquot was taken from each culture flask and serially diluted and 5 $\mu$l was spotted (three technical replicates) on agar plates containing appropriate antibiotics (100 $\mu$g/ml of rifampin or 15 $\mu$g/ml of tetracycline). V52 monocultures were spotted on rifampin plates whereas ETEC E616 and E36 monocultures were spotted on tetracycline plates. Following, all cocultures were spotted on both sets of antibiotic plates to distinguish between the individual strains during cocultures. All plates were incubated for a period of 12 to 16 h at 37°C after which the colonies were counted and the CFU/ml value was calculated.

**DNA extraction, sequencing, and genome assembly.** Genomic DNA and plasmids (in the case of ETEC) were extracted from bacterial cells for the purpose of whole-genome sequencing. *V. cholerae* and ETEC cells (monocultures) were inoculated in rich LB (Sigma-Aldrich) medium and grown at 37°C at 200 rpm until stationary phase. Subsequently, cells were harvested and lysed and the genomic DNA was

extracted using the DNeasy blood and tissue kit (Qiagen), according to manufacturer's instructions. Plasmid DNA from both the ETEC strains was additionally isolated using the Gene Jet plasmid miniprep kit (Thermo Scientific) by following the manufacturer's instructions.

Genome sequences were assembled using SPAdes (84) for *V. cholerae* V52 and SPAdes and plasmid-SPAdes (85) for ETEC E616 and ETEC E36. PATRIC (57) and eggNOG mapper (86) were used for genome annotation.

**Reconstruction of *V. cholerae* GEM *i*AM-Vc960.** A list of metabolic pathways in *V. cholerae* V52 was built based on the genome annotation generated in this study as well as those available in PATRIC and that of *V. cholerae* O1 N16961 (see Table S1 at https://github.com/alyamahmoud/coinfection_modeling/ blob/master/supplementary_material/supplementary_tables.xlsx). The reconstruction was converted into a model, and the stoichiometric matrix was constructed with mass- and charge-balanced reactions in the standard fashion using the COBRA toolbox v.3.0 (33). Flux balance analysis was used to assess network characteristics and perform simulations (34). We used *i*JO1366 (35) as a starting point for reconstruction efforts; it is a common practice to use the closest available species as a starting template (13, 14) while keeping only reactions for which evidence exists of their presence in the *V. cholerae* genome and/or transcriptome (see Table S1 at https://github.com/alyamahmoud/coinfection_modeling/ blob/master/supplementary_material/supplementary_tables.xlsx). We also built an objective biomass function based on *i*JO1366 and *V. vulnificus* (7) previously reconstructed GEMs. Additional reaction content was added from KEGG and BIOCYC databases. All reactions added were manually curated according to a published protocol (83). *i*AM-Vc960 was assessed for mass balance (83). Metabolites charges and formulae were obtained from BiGG (87) and updated in *i*AM-Vc960 to mass-balance the respective reactions. All reconstruction, refinement, validation, and simulations using all models in this study were done using the COBRA toolbox (33) (v3.0.) and Matlab-R2016b. Please refer to section "Refinement of *i*AM-Vc960" in the supplementary text at https://github.com/alyamahmoud/coinfection _modeling/blob/master/supplementary_material/supplementary_text.docx for more details on the curation steps of *i*AM-Vc960.

**Validation of *i*AM-Vc960 single gene deletion essentiality predictions.** We downloaded gene essentiality data for *V. cholerae* O1 strain C6706 from the Online GEne Essentiality (OGEE) database (4, 5). In total, 3,886 genes (total number of ORFs identified in *V. cholerae*) were tested for essentiality. Four hundred fifty-eight genes were essential, 148 were essential for fitness, 3,144 were nonessential, and 136 were unknown. Out of the 458 essential genes, 145 were metabolic genes and were already in *i*AM-Vc960. *i*AM-Vc960 predicted 94 of those to be essential while the remaining 51 were falsely predicted by the model as nonessential. For the nonessential genes, 758 of those were already in *i*AM-Vc960. The model could predict 693 as nonessential while 65 were falsely predicted by the model as essential. The overall accuracy of the model-predicted single gene essentiality was 87% (Fig. 2B). This discrepancy between the model predictions and the high-confidence set that we used earlier, and assuming a low experimental error rate, indicates that the reconstructed *V. cholerae* reactome is incomplete and that there is further room for improvement and refinement of the *i*AM-Vc960, representing opportunities for new biological discoveries.

**Metabolic modeling of coinfection of *V. cholerae* and ETEC.** To simulate coinfection, individual species models were combined into a community model where each species would interact with a common external metabolic environment through their metabolite exchange reactions (55, 56). This allowed each species to access the pool of medium/infection site metabolites as well as metabolites that were released/taken up by the other pathogen. Each species could secrete/take up only those metabolites for which an exchange reaction (e.g., via transporters or free diffusion) exists in the model. The widely employed FBA objective of biomass maximization (34) was replaced with the maximization of a weighted sum of the biomass production fluxes for the community members (88), i.e., the objective function was set to maximize the biomass function of each pathogen, simulating growth at 1:1 species composition/abundance. Flux balance analysis (FBA) was performed using open CORBA in Matlab 2016b and the Gurobi solver v7.0. Please refer to section "Quality control of the coculture model *i*Co-Culture2993" in the supplementary text at https://github.com/alyamahmoud/coinfection_modeling/ blob/master/supplementary_material/supplementary_text.docx for more details on the curation of the coculture model.

**Catabolic capabilities of *V. cholerae*, ETEC, and coinfection GEMs.** Growth under 656 different growth-supporting conditions was simulated for *i*AM-Vc960, *i*ETEC1333, and *i*Co-Culture and then compared to identical simulation conditions for 55 GEMs of *E. coli* and *Shigella* (13). Table S4 at https://github.com/alyamahmoud/coinfection_modeling/blob/master/supplementary_material/ supplementary_tables.xlsx details the simulation conditions for the alternative nutrient sources, and Table S5 at https://github.com/alyamahmoud/coinfection_modeling/blob/master/supplementary _material/supplementary_tables.xlsx shows all simulated growth conditions. Nutrient sources with growth rates above 0.01 were classified as growth supporting, whereas nutrient sources with growth rates less than 0.01 were classified as non-growth supporting. The binary results from the growth/no-growth simulations were used to reconstruct the heatmap (Fig. 3C). Ward's agglomerative clustering of the matrix of correlations was used to cluster the species. The heatmap was visualized using the pheatmap R package. The ternary plot (Fig. 4B) was visualized using the ggtern R package (89).

**RNA extraction, sequencing, and data analysis.** Sampling of cells for the purpose of RNA extraction was performed as follows. Bacterial cells (monocultures and cocultures of *V. cholerae* and ETEC) were grown to mid-logarithmic phase in shake flasks at 37°C at 200 rpm. In the case of the cocultures, equal concentrations of individual monocultures were inoculated into the same medium from the start. Once the appropriate growth phase was reached, the cells were harvested. RNA was

extracted from the harvested cells using the RNeasy minikit (Qiagen), according to manufacturer's instructions. Experiments were carried out in triplicates. The RNA extracted was in the range of 200 to 100 ng/$\mu$l.

RNAseq reads from monocultures were directly aligned to the genome assembly of the corresponding species. To check for reads cross-mapping, we first attempted to map *V. cholerae* reads against the ETEC genome assembly and vice versa. In either case, the percentage of mapped reads was <2% (see Fig. S4 at https://github.com/alyamahmoud/coinfection_modeling/blob/master/supplementary_material/supplementary_text.docx), indicating minimal cross-mapping between the two species. Following, we constructed an artificial genome assembly of both *V. cholerae* and ETEC combined, i.e., representing the coculture as a single entity by merging the genome assemblies of the two species. PATRIC (57) was used for annotation of the merged genome assembly. *V. cholerae* and ETEC reads from the coculture were then each separately aligned against the merged genome assembly, and read counts were computed, i.e., we sequenced and annotated the genome sequences from the single and dual cultures using the same assembly and annotation pipeline to avoid differential gene calling. Although all strains used in this study (*V. cholerae* V52 and ETEC E36 and E616) are clinical isolates that have been sequenced and characterized before (59, 60), we have generated new assemblies and annotations mainly for the sake of consistency for gene calling where we subjected the mono- and coculture transcriptomes to the same processing and annotation pipelines. Bowtie2 (90) was used for all genome alignment. Read counts for all genes were extracted with HTSeq-count (91) and normalized and analyzed using the R package DESeq2 (92). In order to do differential expression analysis between the genome assemblies generated from the monocultures and the cocultures, we aggregated genes by their FIGfam identifiers (IDs) (93). Members of a FIGfam are believed to implement the same function, they are believed to derive from a common ancestor, and they can be globally aligned. We wanted to see if there are specific functions that will be significantly altered between the two culture conditions, especially since the sequence identity between ETEC and *V. cholerae* is around 80% (40). FIGfam IDs were aggregated by keeping the FIGfam ID with the maximum value of raw read counts across all replicates from both the mono- and cocultures. The GOstats (94) R package was used for the GO enrichment analysis, and the GOplot (95) R package was used for visualization of GO enrichment results in Fig. 5. The details of the procedure for dual RNAseq data analysis are outlined in Fig. S4 and in the supplementary text at https://github.com/alyamahmoud/coinfection_modeling/blob/master/supplementary_material/supplementary_text.docx, and code is shown at the GitHub repository at https://github.com/alyamahmoud/coinfection_modeling.

**Data availability.** All data generated in this study are included in this published article. Models, supplementary text, and supplementary tables as well as code to reproduce the main figures and key analyses in this study are available as part of a GitHub repository at https://github.com/alyamahmoud/coinfection_modeling.

## ACKNOWLEDGMENTS

We are very grateful to insightful comments from Nathan Lewis and Neema Jamshidi. We thank Abdallah Abdallah and Mohammed Alarawi from the bioscience core lab at KAUST, Hajime Ohyanagi, and Yoshi Saito for helpful discussions.

This research was supported by funding from KAUST, BAS/1/1624-01-01 and BAS/1/1059-01-01 and SEED funding FCS/1/2448-01-01 (A.M.A.-H., X.G., T.G., and K.M.) as well as by grants from the Novo Nordisk Foundation (I.M. and V.R.), the Swedish National Research Council (V.R.), and the Danish National Research Council (DFF) (to I.M.).

A.M.A.-H. performed the modeling, simulations, and data analysis and wrote the paper. V.R. performed the experiments. B.J. provided support for the modeling and data analysis. J.N. provided support for the modeling. K.M. provided support for the DNA and RNA sequencing. X.G. contributed to data analysis. I.M. and T.G. conceived the project, oversaw the project, and wrote the paper. All authors read and approved the final manuscript.

We declare no competing interests.

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
