## [Reviewer comments · mSystems]

Integrated metabolic modeling, culturing and transcriptomics explains enhanced virulence of *V. cholerae* during co-infection with ETEC

Alyaa Abdel-Haleem, Vaishnavi Ravikumar, Boyang Ji, Katsuhiko Mineta, Xin Gao, Jens Nielsen, Takashi Gojobori, and Ivan Mijakovic

Corresponding Author(s): Ivan Mijakovic, Chalmers University of Technology

Review Timeline:

Submission Date:

June 12, 2020

Accepted:

July 19, 2020

Editor: David Cleary

Reviewer(s): The reviewers have opted to remain anonymous.

Transaction Report:

DOI: <https://doi.org/10.1128/mSystems.00491-20>

Reviewer #1 (Comments for the Author):

The work presented in the following manuscript, entitled 'Integrated metabolic modeling, culturing and transcriptomics explains enhanced virulence of *V. cholerae* during co-infection with ETEC', addresses interesting experimental and clinical problems, uses appropriate computational and experimental tools, and is well-written. I thoroughly enjoyed reading it. The authors generated computational models of the metabolism of *E. coli*, *V. cholerae*, and the two together during coinfection. They compared the predicted metabolic requirements and gene essentiality of each pathogen alone and in coculture. Experimental data supports their computational predictions, and highlight an additional set of (potentially targetable) metabolic weaknesses for *V. cholerae* during coinfection. This work emphasizes (1) the largely unappreciated differences between in vitro and in vivo infection, (2) the role of in silico experimentation to improve the clinical utility of in vitro approaches, and (3) the growing appreciation for the context-dependency of pathogen phenotypes. However, without providing the metabolic network reconstructions with the manuscript, I cannot evaluate a critical component of this work.

Major critiques:

1) Greater clarity is needed regarding the reconstruction and manual curation processes.

a. Please add an overview of the reconstruction approach to the main text, including the software used and relevant assumptions.

We have added lines 172-179 to describe the reconstruction approach and referred the interested reader to more in depth details in the supplementary text and tables.

. . . "A list of metabolic pathways in *V. cholerae* V52 was built based on the genome annotation generated in this study, those available in PATRIC and that of *V. cholerae* O1 N16961 (Table S1). The reconstruction was converted into a model and the stoichiometric matrix was constructed with mass and charge balanced reactions in the standard fashion using the COBRA toolbox v.3.0.¹ Flux balance analysis was used to assess network characteristics and perform simulations². The biomass function was constructed primarily based on that of *V. vulnificus*³ and *E. coli* K12 iJO1366⁴. Transcriptomics data of *V. cholerae* V52 single-cultures in minimal media was also generated and used to further refine iAM-Vc960 reconstruction and biomass objective function (Table S1)."

b. Please document all manual curation steps, bounds for exchange reactions in mono- and co-culture, and parameters for the 656 conditions.

We have included details and examples of manual curation steps for individual reactions in the supplementary text under the section "Refinement of iAM-Vc960". We have directed interested readers to refer to this section in the methods section of the main text:

.. "Please refer to section "Refinement of iAM-Vc960" in the supplementary text for more details on the curation steps of iAM-Vc960."

We added lines to the legend of Figure 3 to point the reader to the details of the simulation conditions:

“Table S4 details the simulation conditions for the alternative nutrient sources and Table S5 shows all simulated growth conditions. A growth rate of 0.01 was used as the cutoff for binarizing the simulation results and was used to construct the heatmap”. Please refer to section “Prediction of all growth-supporting carbon, nitrogen, phosphorus, and sulfur sources” in the supplementary text for details.

We have also included details of the refinement of the co-culture model in the supplementary text and pointed interested readers to refer to this section in the methods section of the main text:

“Please refer to section “Quality control of the co-culture model iVc-ETEC-2293” in the supplementary text for more details on the curation of the co-culture model”.

c. Please discuss in the supplement the sink and demand reactions. Are they necessary for the accuracy of the growth rate, or for the model to function?

We have performed a single-reaction deletion analysis for every reaction in the single species and dual-species model.

iAM-Vc960 has a total of 11 demand and sink reactions. 2 demand reactions (DM_4CRSOL, DM_MTHTHF) and 2 sink reactions (sink_glu1sa[c], sink_glyclt[c]) are necessary for the model to function, the remaining 7 reactions do not affect the accuracy of the growth rate or the model function.

DM_MTHTHF: as per iJO1366; needed to allow (2R,4S)-2-methyl-2,3,3,4-tetrahydroxytetrahydrofuran to leave system. We have made it irreversible to allow leaving only and not uptake)

DM_4CRSOL: as per iJO1366, needed to allow p-Cresol to leave the system. We have made it irreversible to allow leaving system only and not uptake.

sink_glyclt[c]: Glycolate dehydrogenase (EC 1.1.99.14), subunits *glcD/E/F* are not annotated in the *V. cholerae* genome. Hence, reactions GLYCTO2, GLYCTO3, GLYCTO4 were removed. A sink reaction was nevertheless added for cytosolic glycolate, since no literature evidence supported the presence of a glycolate dehydrogenase-encoding gene in the *V. cholerae* genome. However, *glcE* was predicted to be essential for optimal growth.

sink_glu1sa[c]: This sink reaction is added to supply glu1sa[c] for downstream reactions instead of being supplied by glutamyl tRNA reductase reaction in iJO1366. We removed all tRNA reactions since they are beyond the scope of direct metabolic reactions modeled in this study.

We have appended the relevant details to the supplementary text under the section “Refinement of *iAM-Vc960*”

2) The reconstruction must be provided. Additionally, per a recent community effort in the metabolic modeling field (<https://www.biorxiv.org/content/10.1101/700112v3>), I request that the authors use the 'checklist for authors' (Box 2 in the preprint).

The reconstructions are available as part of the supplementary material to the paper and are now also available as part of a github repository: https://github.com/alyamahmoud/coinfection_modeling . As per the reviewer request, we have used the checklist provided in the cited pre-print. Please find our responses below.

Box 2: Proposed minimum standardized content for a metabolic network reconstruction. We propose that modelers use this list as a guide to help standardize accessibility, content, and quality; however, more comprehensive documentation and more interpretable and accessible information can only improve the usability and biological relevance of the shared reconstruction.

Model

- Recognized naming convention
- Machine-readable reference to organism and species embedded via MIRIAM annotation (including other classification(s), if necessary)
- NCBI reference genome
- Author(s) contact information embedded

Metabolite

- Human readable, descriptive name (e.g., D-Glucose)
- Charge (e.g., 0)
- Chemical formula (e.g., C₆H₁₂O₆)
- InChi strings
- At least one database identifier from a reliable resource, such as:
 - MetaNetX
 - BiGG
 - KEGG Compound
 - ChEBI
 - SEED
 - HMDB

Biochemical reaction

- Human readable, descriptive name (e.g., phosphofructokinase)
- Reaction formula
- Stoichiometric coefficients
- At least one database identifier from a reliable resource, such as:
 - MetaNetX
 - BiGG
 - KEGG Reaction
- EC Number

Gene

- Name and/or identifier
- DNA sequence ID
- Protein sequence ID
- Position (including chromosome, if applicable)
- Associated reactions (GPR)

Using the reviewer checklist (Box 3 in the preprint), the following components are missing or unclear:

- a. Is the reconstruction publicly available and shared on an accessible database? (I am happy to evaluate the remaining questions, if provided the model)

All three reconstructions (iAM-Vc960, iETEC_1333 and iVc-ETEC_2293) built and modified in this study are available as supplementary material to this study and as part

of a github repository: https://github.com/alyamahmoud/coinfection_modeling. The models will also be deposited in BiGG when the manuscript has been published.

Is the reconstruction saved in a language-independent format (i.e., SBML)?

Yes, *iAM-Vc960* is available as .SBML. The *iVc-ETEC-2293* is available as a .mat file and the *iETEC-1333* is available as part of a previous publication (PMID: 24277855)

b. The coculture reconstruction name does not indicate species.

We have modified the name of the co-culture reconstruction to *iVc-ETEC-2293* to reflect the species names. However, this is a modular approach where the several other species can replace ETEC and/or *V. cholerae*.

c. Are identifiers (genes, metabolites, reactions) consistently from one namespace?

Yes. All metabolites and reactions are obtained from BiGG. Gene symbols are obtained from the genome annotation of *V. cholerae* O1 biovar El Tor str. N16991 since this is the closest complete genome sequence available for *V. cholerae*.

For the gene-protein-reaction (GPR) building, 28 genes did not have a *V. cholerae* specific

gene ID. If the rule was 'AND' and not all orthologous were found in the *V. cholerae* genome or transcriptome data generated in this study or in Uniprot, we kept the gene symbol to indicate that this specific gene has not been annotated in *V. cholerae*, but it is potentially needed for the corresponding reaction. Also, for this set of reactions, at least one of the genes was expressed according to the transcriptomic data generated in this study. For example, AACPS1 is associated to (*aas* and VC2020), there is proteomic evidence for VC2020 expression in *V. cholerae* according to Uniprot, and it is also expressed in V52 (Table S9-S10). However, there is no evidence for *aas* expression and it does not have a designated VC- or VCA- formatted ID in Uniprot, so the gene symbol

was

used.

We realize that *V. cholerae* V52 and *V. cholerae* N16961 belong to different serotype groups but we make no distinction in relation to serotype groups in this study. We have used *V. cholerae* O1 El Tor N16961 genome assembly as it is the most complete and update genome annotation for *V. cholerae*. We have also sequenced and annotated the genome of *V. cholerae* V52, since it is more frequently co-isolated with ETEC in clinical settings⁵. Future studies will focus on assessing the linkage between metabolic capabilities and serotype groups as well as across the entire species.

d. Have Memote (Lieven et al, 2018) tests been run? (This would obviate the need for answering b, d, f, and h.)

We have attempted to run the released version of memote but it gets stuck at the test_consistency.py step as part of the benchmark testing. However, we have replied in details to the questions b, d, f and h here.

e. Is the objective reaction indicated?

Yes, the objective reaction for each model is indicated (retrieved through the cobra toolbox by find_model.c).

g. Are exchange, sink, and demand reactions and all necessary constraints included as defaults or in code?

They are included as defaults.

h. Can the reconstruction be instantiated without error with COBRA software?

Yes.

i. Are simulation parameters (objective reaction, constraints, etc.) provided in any of the following formats (include at least one): README.md file? COMBINE repository? full analytic code? e.g., iPython notebook or equivalent

The models are packaged with the necessary constraints to reproduce the main findings of the manuscript. We have also included the analytical code as .mat file to reproduce the 600+ tested growth conditions and single gene deletion analyses. In addition, we are now providing an R markdown file to reproduce the main figures in the manuscript.

j. Is the COBRA software version documented (and referenced)?

We have added that to the methods section now.

'...All reconstruction, refinement, validation and simulations using all models in this study were done using the COBRA toolbox (v3.0) (CITATION) and Matlab-R2016b.'

k. Are COBRA software efforts appropriately credited?

We have revised the manuscript and included sentences to indicate whenever COBRA toolbox was used for building the reconstruction, developing the models or running *in silico* simulations.

3) Does the V cholerae transcriptome respond in the same way to both ETEC strains? i.e. if you were to perform PCA on both datasets at once (in other words, plot both V cholerae Fig. S3 subpanels on the same components), how would the co-culture clusters look?

We thank the reviewer for this thought-provoking question. We have performed the PCA plot as per the reviewer's suggestions.

Based on the dual transcriptome analysis, 20% and 17% of *V. cholerae* transcriptome was perturbed when co-cultured with E36 and E616, respectively which does not reflect a significant difference in the 'quantity' of perturbation caused by either ETEC species. Further, as shown in Figure 5, the differentially expressed genes in either co-cultures relative to *V. cholerae* monoculture are overlapping.

Taken together, we are inclined to say that *V. cholerae* seems to respond in the same way to either ETEC strains (contrary to ETEC response to V52 which shows significant difference in terms of quantity and type of perturbation for E616 vs. E36).

4) Lines 234-252 – these conclusions are supported by the data; however, the figure does not make these conclusions. I would suggest the data in both panel B and C be aggregated by genus, rather than presenting each model independently. E.g. boxplots in Panel B and use the color of the heatmap in Panel C to represent the percent of models that could grow.

We have now updated Figure 3 to reflect those changes.

5) Lines 320-328 – are there any conditions in which cross-feeding did NOT occur? I.e. conditions in which one species could grow in monoculture but the coculture could not?

No. This is a technical limitation of setting the objective function of the co-culture model as the maximization of the weighted sum of the objective functions/biomass of the two species in the merged community model. Future experimentation with other potential objective functions for the merged model might alleviate this limitation.

6) I am not convinced by the data presented in Figure 4D because the growth curves in

Figure 4C do not appear to be incomplete. You propose that there is a difference in total growth, however, the findings in Fig. 4D could also be explained by merely a shift in the growth curve. Without demonstrating that all cultures reached stationary phase (i.e. take Figure 4C out to a later timepoint), I am not convinced that there is truly a difference in final CFU.

In response to the reviewer comment, we have reproduced the experiment for 10h (instead of 7h). We have included the cfu/ml count at each sampled time point (sampling interval 1h for 10h) for the mono-cultures and co-cultures and how they compare for each species. The cfu/ml confirms the finding that the growth rate of *V. cholerae* in co-culture with ETEC exceeds that of its mono-culture. In contrast, for both ETEC species tested here, the growth of ETEC in co-culture with *V. cholerae* tends to decrease relative to either mono-cultures. The cfu/ml experiment also show that E616 is less sensitive to growth-suppressive effects mediated by *V. cholerae* relative to E36. We have included cfu/ml counts for time points 0h, 5h and 10h in the main figure (Figure 4) and have included the full stack of sampled points in the supplementary material (Figure S3).

7) I find Figure S3 much more compelling than Figure 5. I do not know how to interpret figure 5 – perhaps this would be alleviated by a figure caption, but it was not in the document I had. I would suggest moving Figure S3 into Figure 5 as a subpanel or to replace Figure 5.

We have included the figure legends as part of the figures during the figure submission process, we hope this will be accessible to the reviewer. In response to the reviewer's comment, we have modified the caption for Figure 5 and appended Figure S3 to be part of the main Figure 5.

8) I would appreciate a discussion of the impact of nonpathogenic *E. coli* (commensal microbes). Wouldn't *V. cholerae* nearly always be in the presence of an *E. coli*? – if true, the impact of your conclusions extends beyond coinfection.

V. cholerae V52 was also observed to be virulent against several other Gram-negative species including the commensal *E. coli* K-12 MG1655 although ETEC was not tested⁶. Future studies should focus on simulating growth of pathogenic and environmental *Vibrio* species with commensal and other pathogen *E. coli* species. Since in this study we do not present enough data that would support such a generalized hypothesis, we prefer to not include that particular discussion point at this time.

Minor critiques:

1) Line 166-167 – 'GPRs could be defined for 72% of all enzymatic reactions' – what about transport reactions?

In Table S1, we have provided the details of all reactions included *iAM-Vc960* as to whether they have gene-protein association as well as level of confidence based on genomic, transcriptomic and proteomic evidence. In total, there are 1569 metabolic and transport reactions that have GPRs, 88 transport reactions and 117 metabolic reactions that have no gene-protein reaction association.

2) Line 184: Please state how the high confidence set of genes were established.

It is the overlapping (intersection set) of single-gene essentiality experiments performed in *V. cholerae* by three independent previously published studies:

Chao, M. C., Pritchard, J. R., Zhang, Y. J., Rubin, E. J., Livny, J., Davis, B. M. *et al.* . High-resolution definition of the *Vibrio cholerae* essential gene set with hidden Markov model-based analyses of transposon-insertion sequencing data. *Nucleic Acids Res* **41**, 9033-9048, doi:10.1093/nar/gkt654 (2013).

Cameron, D. E., Urbach, J. M. & Mekalanos, J. J. A defined transposon mutant library and its use in identifying motility genes in *Vibrio cholerae*. *Proc Natl Acad Sci U S A* **105**, 8736-8741, doi:10.1073/pnas.0803281105 (2008).

Kamp, H. D., Patimalla-Dipali, B., Lazinski, D. W., Wallace-Gadsden, F. & Camilli, A. Genefitness landscapes of *Vibrio cholerae* at important stages of its life cycle. *PLoS Pathog* **9**, e1003800, doi:10.1371/journal.ppat.1003800 (2013).

Table S2 details the genes included in this high-confidence set (denoted as E for each of the studies where these genes were reported as being essential).

3) Lines 223-232 – please clarify which references are for experimental data.

We have modified these lines which now read:

“The agreement between the experimental gene essentiality data, obtained from previously published studies, and the computational results, generated in the current study, in terms of growth and single gene essentiality predictions, on the whole, validates the content of the reconstruction, the modeling procedure and the objective function definition (Figure 1, step 1).”

4) Lines 254-257 – Including examples would make this section more compelling.

We have included examples for these in lines 260-270 which reads:

“V. cholerae model completely lost the capability to sustain growth on nutrient sources for which most of E. coli and Shigella models had growth capabilities. Some of these nutrients include D-lactate, D-fumarate, lactose, L-alanine-glutamate, uridine, xanthosine, thymidine, R-Glycerate, sn-Glycero-3-phosphoethanolamine, 4-Hydroxy-L-threonine, L-Asparagine, L-proline, L-Arabinose, and L-Xylulose as carbon sources as well as nitrate, nitrite⁷, ornithine, L-proline, agmatine, uracil, and putrescine⁸, as nitrogen sources, and myo-Inositol-hexakisphosphate as phosphorus sources”

5) Please address whether or not there is additivity in the severity of coinfection. This is alluded to in lines 271-276.

Based on the *in vitro* experiments supported by model predictions and explanation as well as observed increased in antibody titer against cholera toxin in patients with dual infections of *V. cholerae* and ETEC, we believe there is an increased severity in cholera infection when ETEC is co-existing. However, there is no observed increase in ETEC severity. Furthermore, there is increase in the severity of diarrhea as reported in Chowdhury *et al.*, 2010⁵.

6) Line 308 – I understand crossfeeding to be the direct transfer of a metabolite (e.g. Organism 1 produces metabolite A and organism 2 consumes metabolite A, so A is crossfed). In this text, it seems like you are describing metabolites as the ‘crossfed metabolite’ if they create an environment that supports the direct transfer of a metabolite (e.g. Organism 1 consumes X and then produces metabolite A, and organism 2 consumes metabolite A, so X is crossfed). Please clarify or define to correct the readers preconceived definition.

Yes, indeed we refer to the second definition as provided by the reviewer here. We have added those lines to the legend of Figure 4 to assert the definition of cross-feeding we are assuming here.

7) This statement is confusing: ‘There are several gene deletions associated with reactions for which drugs have not been developed’ (lines 437-438). Are you saying that there are several reactions that could be targeted?

Yes, indeed. We meant to highlight that among the list of essential genes for growth of *V. cholerae* in both infection scenarios (mono- and co-cultures), some of them had evidence to support their essentiality (by having drug inhibitors already designed to block their associated reactions). However, the majority of those predicted essential targets (genes) are novel and have not been taken into clinical testing.

8) Figure 2: what is ‘MCC’?

Mathew correlation coefficient. It is used for estimating model performance when there is class imbalance mainly. We have indicated what the acronym MCC denotes in the figure legend.

9) Is there a break in the scale of the bottom of Figure 4D? Please clarify.

We have revised Figure 4 as per the reviewer’s recommendations and Figure 4D now has the growth culture OD measurements. Figure 4C (was previously 4D) is now revised and contains no breaks.

10) Some experimental details should be moved to the main text for clarity. For example, please briefly explain the experimental approach for generating Fig. 4D and describe the experimental coculture setup in the main text.

We have added Lines 359-372 to the main text to detail the co-culture setup.

*“To this end, we developed a robust in vitro co-culture system of *V. cholerae* V52 and two different ETEC strains (E36 or E616) in M9 minimal medium supplemented with glucose (Figure 4C-D). All three tested strains (V52, E36 & E616) are clinical isolates that have been sequenced and characterized before^{62,63} (see supplementary text for details on strain selection and sequencing performed as part of the current study). We determined the impact of the co-culture on each strain’s growth by comparing single culture abundance after 10 hours of growth to the abundance of each strain in co-culture at the same time (determined using CFU counting; all strains were in transition or stationary phase, Figure S3). Simultaneously, at every hour, an aliquot was taken from each culture flask, serially diluted and 5µL were spotted (three technical*

replicates) on agar plates containing 100µg/mL of rifampicin or 15µg/mL of tetracycline for V52 and ETEC, respectively. All co-cultures were then spotted on both sets of antibiotic plates to distinguish between the individual strains during co-cultures. E36 and E616 were shown to have diminished ability to grow in co-culture with *V. cholerae* V52. By contrast, growth of *V. cholerae* V52 was strongly enhanced in co-culture conditions (Fig. 4C-D).”

11) Figure 6 - It would be interesting to see the false positive and false negative predictions in different colors. Additionally, were there no essential genes in only the co-culture condition? If so, this is interesting – please discuss.

While we thank the reviewer for this comment, we believe that the map is already showing a lot of information, and we would like it to highlight the targets where essentiality is conserved between the mono and co-cultures while also highlighting that the model predictions are not generally perfect. There were no predicted cases of acquired essentiality in co-culture relative to mono-culture conditions of *V. cholerae*. This can be due to the experimental setup implemented here being based on maximizing the additive growth of the two species in co-culture and adopting the model- and *in vitro*-experiments supported explanation that the observed increase in *V. cholerae* growth is mediated by cross feeding.

Typo/copy editing suggestions:

Line 52: Enterotoxigenic is italicized.

Line 74: *in vitro* should be italicized.

Line 77: ‘Genome’ should be ‘transcriptome.’

Line 129: New paragraph break before ‘While most case...’

Line 153: Extra text between ‘xxx’s

Lines 314-317: parentheses are unnecessary.

Line 537: ‘miss out on’ is a little casual.

Line 710-711: extra closing parenthesis between DFF and IM

Author contributions: two groups of people are credited for the project idea (AM line 714, IM and TG line 717)

We thank the reviewer for the detailed review. We have fixed all the typo/copy editing suggestions as per the reviewer recommendations.

Reviewer #2 (Comments for the Author):

This work is certainly impressive and handles a computationally and biologically elaborate question, which is the competition/cooperation between two pathogenic strains. Of course, growth on M9 is not the ideal habitat for those two human parasites; however, it is a first step to understand their behavior when co-cultured.

The study has three major points of strength: 1) establishing a robust *in vitro* co-culture model, allowing the separate estimation of each microbial density (based on antibiotic selection); 2) establishing a co-infection GEM; 3) RNA-Seq of both pathogens when co-cultured, which offers interpretation of some of the observed co-dependencies.

The work is technically sound and I have no major concerns.

My main training/background is in genomics and microbial pathogenesis, thus I am not the best person to judge the extent of innovation in this work. I understand that communities of two or even three microbes have been modeled before, but I cannot evaluate the claim that this is "the first study to investigate a synthetic enteric pathogens community" HOWEVER, I would always discourage any authors from writing such marketing-style phrases, which are banned by some top-tier journals. Everybody wants to be the first; however, it's a useless claim, in my opinion.

We thank the reviewer for this suggestion and we completely agree. We have revised the manuscript for any instances where this notion was used.

The manuscript is clearly written, in general, but suffers some minor language/punctuation sins which I'll highlight in a following paragraph*.

Other than that, my only concerns that need to be addressed are:

1) The codes HAVE TO be deposited in a code repository. "available upon request" is no longer acceptable as good practice. Research teams are not eternal, and things get lost when primary authors or lab directors move around.

In response to the reviewer's comments, we have created a github repository that contains all code and intermediate data files to generate the figures in the main text and reproduce the analysis. We have also included the models with default constraints which enables reproducing all simulated growth conditions used in the manuscript. The github repository is available at : https://github.com/alyamahmoud/coinfection_modeling

2) The authors are generous and transparent with supplementary data; however, some tables are too raw to be used. Specifically, Table S5, which is pivotal to understanding the paper and verifying all claim about growth co-dependence/ cross-feeding, is really hard to interpret. Since the table is provided as a tab-delimited one, it's hard to know what fields to focus on. The third row (iCo-culture) is quite important, but some legend, or else a couple of rows of comments, could help whoever wants to understand and verify. What is "highlighted in blue"? This applies to other details in other tables.

In response to the reviewer's comments, we have provided the supplementary tables in excel format at the github repository associated with this manuscript at https://github.com/alyamahmoud/coinfection_modeling. We have also edited the legend for the tables to be more descriptive of the content.

3) In several instances, there's a mixup between sections. Some results are listed under Methods and some discussion is included under Results. Specifically, Lines 628-632 are all results and need to be moved or removed (if redundant). Lines 254-257 are all Discussion material and thus should be moved to that section.

We have revised the entire manuscript for these instances. We hope the current version reads better.

4) Why 0.5% glucose? It's often 0.2% or 0.4%? Not that the concentration matters-since it's the same with all-but I'm curious.

We have opted to use 0.5% as it is a standard concentration used in microbiology and several key references on *V. cholerae* report using this glucose concentration^{9,10}, so this choice was made to allow best possible comparison to existing studies.

* Now below are the language/style issues:

-- GENREs? I'm not sure what is the need of this abbreviation since it's just mentioned once (and honestly it's a really bad sort-of-acronym. GEMs is widely used and accepted as an abbreviation now, but I feel "Genome-scale metabolic network reconstructions" is better not abbreviated, since the abbreviation is only repeated in one paragraph.

We have omitted GENREs and used GEMs as per the reviewer's recommendation.

-- Line 59 (Abstract): "decrease in ETEC growth": not a very accurate expression. The term "decrease" would better qualify "rate" or "extent" of growth, or both.

-- "The majority of studies" is verbose/jargon: "Most studies" is better (Line 65 and 91/ also 298 and elsewhere)

Corrected.

-- In vitro should be in italics everywhere. Sometimes it's not (e.g., Line 74).

Corrected.

-- The use of tense need a quick editorial revision. The use of past simple should often replace the present perfect tense (e.g., "we have used" in Line 70 and "have investigated" Line 486)

Corrected.

-- Line 71: "Enterotoxigenic E. coli" no need to capitalize "enterotoxigenic" here or elsewhere unless a first word of a sentence.

Corrected.

-- Line 90: "despite the fact that pathogens" a textbook example of jargon! It should simply read: "although pathogens"

Corrected.

-- Line 100: "also" and "as well" in the same sentence are redundant. I would remove "also"

Corrected.

-- Line 171: missing "to" after "unable"

Corrected.

-- Line 236: "pathogen E. coli"  pathogenic
Corrected.

-- Line 531: remove the comma after experiments. The rule is not to separate a subject from its verb with a single comma. It needs revision in the entire manuscript.
Corrected the particular one after experiments.

-- Line 559: "this is the first study"  a useless phrase. This is a great study and doesn't need this sentence as there are already co-culture models, so being the first "enteric pathogen" or not is not the issue. Some journals ban such expressions.
Corrected

-- Line 622: "458 genes" Spell out in letters or avoid starting with a number.
Corrected.

-- Lines 687-689: all punctuation here needs revision.
Corrected.

Reviewer #3 (Comments for the Author):

We have read "Integrated metabolic modeling, culturing and transcriptomics explains enhanced virulence of *V. cholerae* during co-infection with ETEC" by Abdel-Haleem et al. The authors developed and validated a new genome-scale model of *V. cholerae*. This was used to simulate co-infections with Toxigenic *E. coli* strains as well as to identify essential genes in co-infections that could be used to design drugs against cholera infections. They then combined this model with a model of ETEC to simulate co-infection in over 600 computational environments. The results suggest that ETEC co-infections enable *V. cholerae* to grow in more environments than they would alone. The gene essentiality predictions are also altered when grown in co-infection with ETEC strains. The study is interesting and tackles an interesting problem. Overall these results are very interesting, but the manuscript would benefit from additional discussion on some key technical details as well as a more thorough proofreading.

Major comments

1. In genome-scale models both gene-essentiality and growth rate predictions are heavily dependent on the formulation of the biomass equation (objective function). However, the authors don't discuss how they formulated the biomass equation for their new iAM-Vc960 model. This must be discussed in the results section on characterizing and building the model.

We agree with the reviewer's comment. We have added lines to the main text to describe the general approach for formulating the biomass function:

*"The biomass function was constructed primarily based on that of *V. vulnificus*³ and *E. coli* K12 iJO1366⁴. Transcriptomics data of *V. cholerae* V52 single-cultures in minimal media was also generated and used to further refine iAM-Vc960 reconstruction and biomass objective function (Table S1)."*

and how it was validated based on single-gene essentiality predictions vs experimental KO data as well as predicted-growth rate vs experimentally observed growth rate of *V. cholerae* previously reported in literature:

2. Stating that *V. cholerae* "lost" metabolic capabilities may be incorrect. Is there any indication that other strains of *V. cholerae* can metabolize the compounds "lost" or are these capabilities that this species cannot utilize in general? A future study could compare catabolic capabilities amongst different strains of *V. cholerae* to answer this question but for now modifying the wording here or finding a reference to support this statement seems appropriate.

We thank the reviewer for this thoughtful comment and we have opted to revise the wording. Indeed, future experiments would assess variations among *Vibrio* species (both pathogenic and environmental) and how their catabolic capabilities might differ.

3. The authors describe that model predictions show enhanced growth for both species in co-infection models. Is this the case in all growth environments, particularly glucose? I understand co-infections enabling growth in new environments however I am confused how co-infection enables enhanced growth in environments like M9-glucose. The *in-vitro* experiment uses only glucose to test this hypothesis. Is there a reason the authors chose this instead of a novel environment that was only capable of growth during co-infection?

The *in vitro* co-growth experiments done in glucose show that only *V. cholerae* growth is enhanced in co-culture conditions while that of ETEC is not. We have then simulated co-infection growth in 656 growth conditions that use alternate carbon (other than glucose), nitrogen, sulfur and phosphorous sources. Among those, *V. cholerae* was able to grow in 51% (n = 336) of the simulated growth conditions, while *E. coli* and *Shigella* were able to grow, on average, in 92% (n = 602) and 75% (n = 493). We thus hypothesize that conditions which enable *V. cholerae* growth in co-infection but not in mono-infection scenarios as well as those that increase *V. cholerae* growth rate are providing essential nutrients to *V. cholerae* in a form it can utilize/catabolize, by being cross-fed through ETEC. However, we are inclined to rely only on findings that come from integration of several lines of evidence, thus further *in vitro* experiments are need to validate the predicted *in silico* growth properties of *V. cholerae*.

4. Beyond these comments there are several minor grammatical and editing errors some of which are highlighted below. We encourage the authors to carefully read the manuscript. Errors with errant "xxx" in the text are particularly worrying.

Minor comments

The first line of the abstract "co-infecting microorganisms alter pathogen gene essentiality during polymicrobial infections" is confusing. Maybe instead say "conditionally essential genes are altered during co-infections of microorganisms"

Corrected.

Line 77 - the genome is not distinct - this should be "transcriptome" - also "were" not "where perturbed". Again on line 375. And 516

Corrected genome to transcriptome.

Line 90 - "essential genes" not genomes

Corrected.

While reading I wondered how often *E. coli* and cholera are co-infections, this is stated in the introduction, but it might be better to add in the importance section.

We have now added this information in the importance section as well:

"In this study, we used genome-scale metabolic models (GEMs) to interrogate the growth capabilities of Vibrio cholerae (V. cholerae) in single and co-infections with enterotoxigenic E. coli (ETEC) which co-occur in a large fraction of diarrheagenic patients."

"xxx" on line 153

Corrected.

- 1 Heirendt, L., Arreckx, S., Pfau, T., Mendoza, S. N., Richelle, A., Heinken, A. *et al.* . Creation and analysis of biochemical constraint-based models using the COBRA Toolbox v.3.0. *Nat Protoc* **14**, 639-702, doi:10.1038/s41596-018-0098-2 (2019).
- 2 Orth, J. D., Thiele, I. & Palsson, B. O. What is flux balance analysis? *Nat Biotechnol* **28**, 245-248, doi:10.1038/nbt.1614 (2010).
- 3 Kim, H. U., Kim, S. Y., Jeong, H., Kim, T. Y., Kim, J. J., Choy, H. E. *et al.* . Integrative genome-scale metabolic analysis of *Vibrio vulnificus* for drug targeting and discovery. *Mol Syst Biol* **7**, 460, doi:10.1038/msb.2010.115 (2011).
- 4 Orth, J. D., Conrad, T. M., Na, J., Lerman, J. A., Nam, H., Feist, A. M. *et al.* . A comprehensive genome-scale reconstruction of *Escherichia coli* metabolism--2011. *Mol Syst Biol* **7**, 535, doi:10.1038/msb.2011.65 (2011).
- 5 Chowdhury, F., Begum, Y. A., Alam, M. M., Khan, A. I., Ahmed, T., Bhuiyan, M. S. *et al.* . Concomitant enterotoxigenic *Escherichia coli* infection induces increased immune responses to *Vibrio cholerae* O1 antigens in patients with cholera in Bangladesh. *Infect Immun* **78**, 2117-2124, doi:10.1128/IAI.01426-09 (2010).
- 6 MacIntyre, D. L., Miyata, S. T., Kitaoka, M. & Pukatzki, S. The *Vibrio cholerae* type VI secretion system displays antimicrobial properties. *Proc Natl Acad Sci U S A* **107**, 19520-19524, doi:10.1073/pnas.1012931107 (2010).
- 7 Bueno, E., Sit, B., Waldor, M. K. & Cava, F. Anaerobic nitrate reduction divergently governs population expansion of the enteropathogen *Vibrio cholerae*. *Nat Microbiol* **3**, 1346-1353, doi:10.1038/s41564-018-0253-0 (2018).
- 8 Beyhan, S., Tischler, A. D., Camilli, A. & Yildiz, F. H. Transcriptome and phenotypic responses of *Vibrio cholerae* to increased cyclic di-GMP level. *J Bacteriol* **188**, 3600-3613, doi:10.1128/JB.188.10.3600-3613.2006 (2006).
- 9 Houot, L. & Watnick, P. I. A novel role for enzyme I of the *Vibrio cholerae* phosphoenolpyruvate phosphotransferase system in regulation of growth in a biofilm. *J Bacteriol* **190**, 311-320, doi:10.1128/JB.01410-07 (2008).

- 10 Lee, D., Kim, E. J., Baek, Y., Lee, J., Yoon, Y., Nair, G. B. *et al.* . Alterations in glucose metabolism in *Vibrio cholerae* serogroup O1 El Tor biotype strains. *Sci Rep* **10**, 308, doi:10.1038/s41598-019-57093-4 (2020).
- 11 Minato, Y., Fassio, S. R., Wolfe, A. J. & Hase, C. C. Central metabolism controls transcription of a virulence gene regulator in *Vibrio cholerae*. *Microbiology* **159**, 792-802, doi:10.1099/mic.0.064865-0 (2013).
- 12 Miller, V. L. & Mekalanos, J. J. A novel suicide vector and its use in construction of insertion mutations: osmoregulation of outer membrane proteins and virulence determinants in *Vibrio cholerae* requires *toxR*. *J Bacteriol* **170**, 2575-2583 (1988).
- 13 Patra, T., Koley, H., Ramamurthy, T., Ghose, A. C. & Nandy, R. K. The Entner-Doudoroff pathway is obligatory for gluconate utilization and contributes to the pathogenicity of *Vibrio cholerae*. *J Bacteriol* **194**, 3377-3385, doi:10.1128/JB.06379-11 (2012).
- 14 Soler-Bistue, A., Mondotte, J. A., Bland, M. J., Val, M. E., Saleh, M. C. & Mazel, D. Genomic location of the major ribosomal protein gene locus determines *Vibrio cholerae* global growth and infectivity. *PLoS Genet* **11**, e1005156, doi:10.1371/journal.pgen.1005156 (2015).

July 19, 2020

Prof. Ivan Mijakovic
Chalmers University of Technology
Department of Biology and Biological Engineering
Kemivägen 10
Goteborg SE-412 96
Sweden

Re: mSystems00491-20 (Integrated metabolic modeling, culturing and transcriptomics explains enhanced virulence of *V. cholerae* during co-infection with ETEC)

Dear Prof. Ivan Mijakovic:

Thank you for the extensive, and extremely thorough responses to the reviewers request. I apologise for the significant delay in coming to a decision, however I am pleased to tell you that your manuscript has been accepted, and I am forwarding it to the ASM Journals Department for publication.

For your reference, ASM Journals' address is given below. Before it can be scheduled for publication, your manuscript will be checked by the mSystems senior production editor, Ellie Ghatineh, to make sure that all elements meet the technical requirements for publication. She will contact you if anything needs to be revised before copyediting and production can begin. Otherwise, you will be notified when your proofs are ready to be viewed.

Sincerely,

David Cleary
Editor, mSystems

Journals Department
Phone: 1-202-942-9338